# The conserved RNA helicase YTHDC2 regulates the transition from proliferation to differentiation in the germline

Alexis S Bailey[1†], Pedro J Batista[2†‡], Rebecca S Gold[1], Y Grace Chen[2], Dirk G de Rooij[3], Howard Y Chang[2], Margaret T Fuller[1]*

[1]Department of Developmental Biology, Stanford University School of Medicine, Stanford, United States; [2]Center for Personal Dynamic Regulomes, Stanford University School of Medicine, Stanford, United States; [3]Center for Reproductive Medicine, Academic Medical Center, University of Amsterdam, Amsterdam, Netherlands

**Abstract** The switch from mitosis to meiosis is the key event marking onset of differentiation in the germline stem cell lineage. In *Drosophila*, the translational repressor Bgcn is required for spermatogonia to stop mitosis and transition to meiotic prophase and the spermatocyte state. Here we show that the mammalian Bgcn homolog YTHDC2 facilitates a clean switch from mitosis to meiosis in mouse germ cells, revealing a conserved role for YTHDC2 in this critical cell fate transition. YTHDC2-deficient male germ cells enter meiosis but have a mixed identity, maintaining expression of Cyclin A2 and failing to properly express many meiotic markers. Instead of continuing through meiotic prophase, the cells attempt an abnormal mitotic-like division and die. YTHDC2 binds multiple transcripts including *Ccna2* and other mitotic transcripts, binds specific piRNA precursors, and interacts with RNA granule components, suggesting that proper progression of germ cells through meiosis is licensed by YTHDC2 through post-transcriptional regulation.

DOI: https://doi.org/10.7554/eLife.26116.001

*For correspondence:
mtfuller@stanford.edu

†These authors contributed equally to this work

Present address: ‡Laboratory of Cell Biology, Center for Cancer Research, National Cancer Institute (NIH), Bethesda, United States

## Introduction

In many adult stem cell lineages that maintain and repair tissues throughout the body, progenitor cells go through multiple transit amplifying divisions before initiating terminal differentiation. To properly execute a clean transition from proliferation to differentiation, cells must both efficiently shut down the preexisting proliferative program and properly initiate the differentiation program. Too little proliferation could lead to tissue degeneration or defective tissue repair, whereas too many divisions may facilitate progression to a tumorous state. Likewise, during embryonic development, cells must make clear decisions to escape a preexisting state and initiate a new program, often within one cell generation.

In the male and female germ cell lineages, germ cells undergo multiple mitotic transit amplifying divisions before switching to the specialized cell cycle of meiosis that leads to terminal differentiation into sperm or eggs (*Kimble, 2011*; *Pepling and Spradling, 1998*). The transition from mitosis to meiosis in the germline is critical for the generation of haploid spermatozoa or oocytes throughout reproductive life. In the *Drosophila* male germline, the DExH-box RNA helicase Benign gonial cell neoplasm (Bgcn) is required cell autonomously for mitotically dividing spermatogonia to stop proliferating and initiate meiosis and spermatocyte differentiation (*Gönczy et al., 1997*). Male germ cells mutant for either *bgcn* or its binding partners *bag-of-marbles* (*bam*) or *tumorous testis* (*tut*) fail to enter meiosis and instead go through several extra rounds of mitosis before they eventually die

(*Chen et al., 2014*; *Gönczy et al., 1997*; *McKearin and Spradling, 1990*). Bgcn also plays an important role in the *Drosophila* female germline, but at an earlier stage of germ cell development, the switch from germline stem cell to transit amplifying oogonial cell. Loss of function of *bgcn* or *bam* results in ovariole tumors composed of stem cell-like cells (*McKearin and Ohlstein, 1995*; *Ohlstein et al., 2000*).

*Drosophila* Bgcn and Bam regulate germ cell differentiation through post-transcriptional control, but with different accessory proteins and different target mRNAs in male versus female germ cells. In the male germline, Bgcn and Bam form a complex with the RNA-binding protein Tut and translationally repress *meiotic protein-26* (*mei-P26*) by binding the 3' UTR of *mei-P26* mRNA (*Chen et al., 2014*; *Insco et al., 2012*). In the female germline, Bam and Bgcn form a complex with Mei-P26 protein and the female-specific RNA-binding protein Sex-lethal (Sxl) in the cystoblast to translationally repress *nanos* (*nos*) mRNA via the *nos* 3' UTR and promote differentiation (*Chau et al., 2012*; *Li et al., 2009*; *Li et al., 2013b*).

Here, we identify the mammalian homolog of the *Drosophila* RNA helicase Bgcn as YTHDC2 and show that it has a conserved, functional role as a critical regulator of the transition from mitosis to meiosis in the mouse germline. Similar to *bgcn*$^{-/-}$ flies, *Ythdc2* mutant male and female mice are viable but infertile. In mouse, both male and female *Ythdc2*$^{-/-}$ germ cells show defects soon after the mitosis to meiosis transition. In *Ythdc2*$^{-/-}$ testes, germ cells attempt to enter meiotic prophase, but fail to properly express many meiotic markers, continue to express Cyclin A2, rapidly condense their chromosomes and initiate an aberrant mitotic-like division before undergoing apoptosis. Analysis of wild-type postnatal testes at the stage when the first wave of germ cells initiate meiotic prophase revealed that YTHDC2 binds a number of mitotic cell cycle RNAs, suggesting that YTHDC2 may play a direct role in turning off the mitotic proliferation program. In addition, YTHDC2 binds specific piRNA precursors and multiple RNAs necessary for terminal differentiation. Consistent with the presence of a YTH domain, 76% of the RNAs bound by YTHDC2 were also enriched in the N6-methyadenosine (m$^6$A) modified fraction of RNAs. Our studies indicate that YTHDC2 facilitates a clean transition from one cell state to the next and suggest that a deeply conserved post-transcriptional control mechanism underlies the germ cell-specific switch from mitosis to meiosis in metazoans.

## Results

### Identification of the mammalian homolog of *Drosophila* Bgcn

Analysis by tBlastn of the nr/nt mouse and human databases using the *Drosophila* Bgcn protein sequence identified mouse YTHDC2 (E-value = 9e$^{-78}$) and human YTHDC2 (E-value = 2e$^{-61}$) as mammalian homologs (*Figure 1A*). The next closest predicted mouse protein was DEAH box polypeptide 36 (DHX36), which was much less similar to *Drosophila* Bgcn (E-value = 1e$^{-30}$) than YTHDC2. Mouse YTHDC2, like *Drosophila* Bgcn, is a member of the DEAD/DExH box family of RNA helicases and shares with *Drosophila* Bgcn the signature DExH RNA helicase domain, a helicase-associated domain (HA2), and several other protein domains, having approximately 44–55% sequence similarity to *Drosophila* Bgcn within each domain (*Figure 1B*). The YTH domain, present in the mouse and other vertebrate homologs, was not recognized in *Drosophila* Bgcn.

Similar to *Drosophila* Bgcn, mouse YTHDC2 is expressed in adult testes. Western blot analysis of adult mouse tissues detected YTHDC2 protein in testis but not in several other adult tissues examined (*Figure 1C*). Immunofluorescence staining of cross-sections of adult mouse testis tubules revealed YTHDC2 in a subset of germ cells in a majority of the seminiferous tubules (*Figure 1D*), where the protein localized primarily to the cytoplasm (*Figure 1E*), similar to *Drosophila* Bgcn. YTHDC2 was not detected by immunofluorescence in spermatogonia next to the tubule wall but was strongly expressed in spermatocytes that co-expressed the meiotic marker SYCP3 (*Figure 1D–F*), suggesting that mouse YTHDC2 is up-regulated in germ cells upon entry into meiosis. Although faint immunofluorescence signal was also detected in round and elongating spermatids in cross-sections of tubules stained with anti-YTHDC2, similar faint fluorescence was also often observed in controls stained only with secondary antibody, suggesting background. The strong signal visible in interstitial cells between testis tubules (*Figure 1D*) was also present in sections of testis from *Ythdc2* null mutants stained with YTHDC2 antibody, indicating that the staining in interstitial cells is non-specific. Staining cross-sections of human testis tubules with anti-YTHDC2 revealed an expression

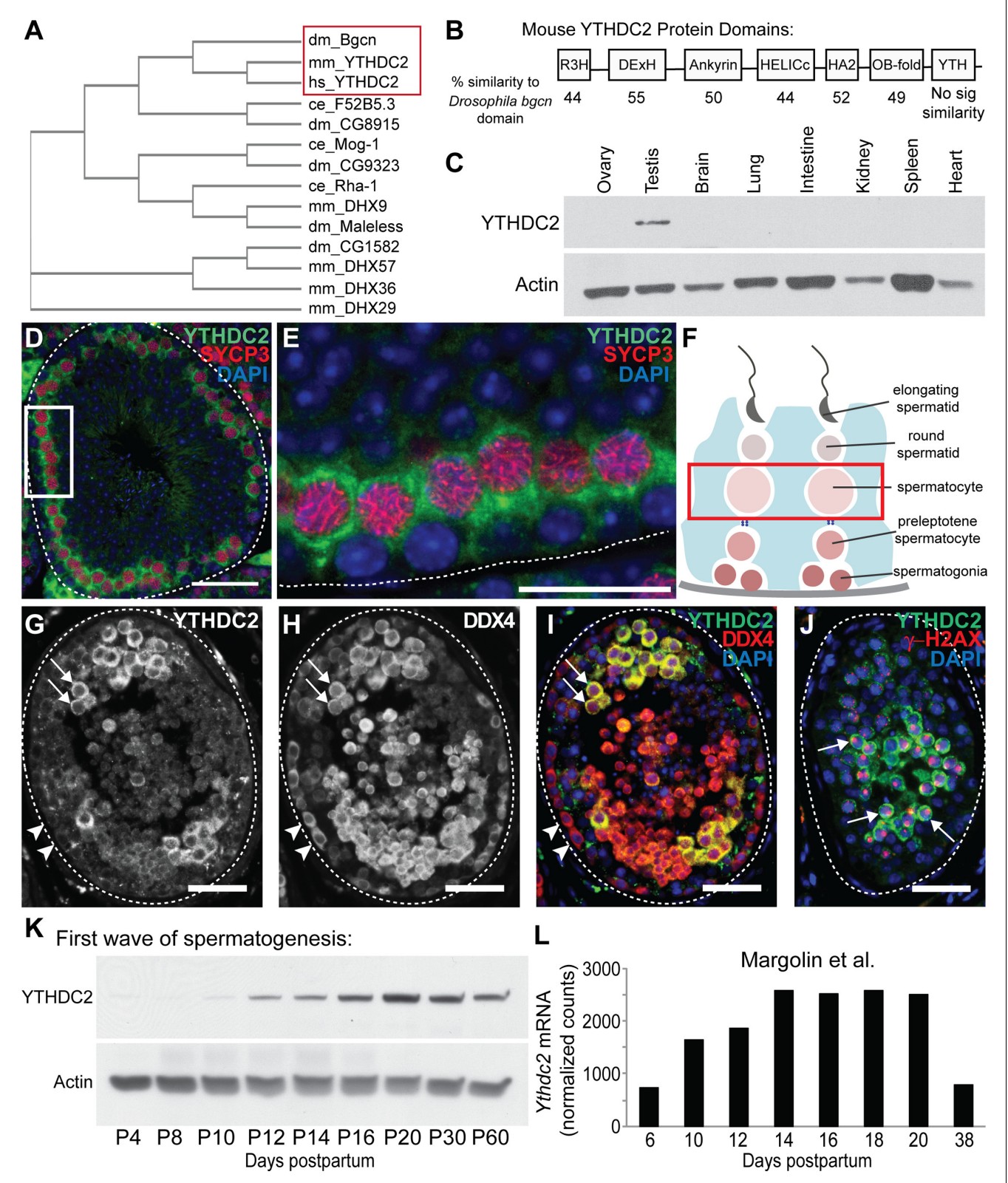

**Figure 1.** A DExH box helicase conserved from *Drosophila* to mammals is expressed in male germ cells entering meiosis. (**A**) Cladogram of Bgcn-related proteins in *Drosophila melanogaster* (dm), *Mus musculus* (mm), *Homo sapiens* (hs) and *Caenorhabditis elegans* (ce). Protein sequences from the NCBI protein database. (**B**) Mouse YTHDC2 protein domains, with percent similarity between mouse YTHDC2 and the corresponding domain of *Drosophila* Bgcn for each domain region. (**C**) Expression of YTHDC2 protein in adult mouse tissues. Western blots probed with α-YTHDC2 or α-Actin.
*Figure 1 continued on next page*

Figure 1 continued

(D) Immunofluorescence images of adult wild-type mouse testis tubule cross-section stained for YTHDC2 (green), SYCP3 (red) and DAPI (blue). (E) High magnification of boxed region in panel D. White dotted line: outline of testis tubule. (F) Diagram of cell stages of spermatogenesis in the testis tubule. Red box: spermatocytes, which express high levels of YTHDC2 protein. (G–I) Immunofluorescence images of adult human testis tubule cross-section stained for (G) YTHDC2 and (H) DDX4. (I) Merged image. Arrowheads: YTHDC2-negative germ cells along tubule periphery. Arrows: YTHDC2-positive germ cells. White dotted line: outline of testis tubule. (J) Adult human testis tubule stained for YTHDC2 (green), γ-H2AX (red), and DAPI (blue). Arrows: YTHDC2-positive pachytene spermatocyte. (K) Expression of YTHDC2 protein in whole testis samples from mice during the first wave of spermatogenesis. Western blots probed with α-YTHDC2 or α-Actin. P, Postnatal. (L) Normalized *Ythdc2* RNA expression during the first wave of spermatogenesis from (*Margolin et al., 2014*). Scale bars: (D and G–J) 50 μm; (E) 25 μm. See also *Figure 1—figure supplement 1*.
DOI: https://doi.org/10.7554/eLife.26116.002

The following figure supplement is available for figure 1:

**Figure supplement 1.** YTHDC2 expression during the first wave of spermatogenesis.
DOI: https://doi.org/10.7554/eLife.26116.003

pattern similar to that observed in mouse, with human YTHDC2 not detected in germ cells around the periphery of the testis tubule (*Figure 1G–I*, arrowheads), but present in the cytoplasm of a sub-set of DDX4-positive germ cells positioned further inward, including pachytene spermatocytes marked by staining of the sex vesicle by γ-H2AX (*Figure 1J*, arrows).

Consistent with the expression pattern of both mouse and human YTHDC2 in the adult testis, analysis of mouse YTHDC2 protein levels during the first wave of spermatogenesis in testes from juvenile males by both western blot (*Figure 1K*) and immunofluorescence staining (*Figure 1—figure supplement 1A–C′*) revealed that expression of YTHDC2 increased between postnatal day 10 to day 12 (P10-P12), the stage when the first wave of early germ cells initiate meiotic prophase. Likewise, analysis of published RNA-Seq data from the first wave of spermatogenesis (*Margolin et al., 2014*) showed that *Ythdc2* RNA levels started to increase at P10 and peaked at P14, when pachytene spermatocytes are present (*Figure 1L* and *Figure 1—figure supplement 1D*).

## YTHDC2-deficient male germ cells fail to properly execute meiotic prophase

We generated two different mutant alleles of mouse *Ythdc2* on a C57BL/6 background: a targeted trap allele (*Ythdc2^tp^*) and a null allele (*Ythdc2^-^*) made by excision of exons 5 and 6 (*Figure 2—figure supplement 1A*). YTHDC2 protein was detected by western blot in extracts of testes from either wild-type or *Ythdc2^+/tp^* mice, but not from males homozygous mutant for either allele (*Figure 2—figure supplement 1E*). Similarly, no YTHDC2 protein expression was detected by immunofluorescence staining of cross-sections of testis tubules from P14 *Ythdc2^-/-^* mice (*Figure 2—figure supplement 1G*). Similar to *bgcn^-/-^* flies, mice homozygous mutant for either the *Ythdc2* targeted trap allele or the null allele were viable, born at the expected Mendelian ratio (*Supplementary file 1*) and developed normally, except that *Ythdc2^-/-^* female mice gained weight more rapidly than wild-type controls (*Figure 2—figure supplement 2A,B*).

Adult male *Ythdc2* homozygous mutant mice were infertile (*Supplementary file 2*), had small testes that weighed significantly less compared to wild-type littermate controls (*Figure 2A–C*) and completely lacked mature spermatozoa in the epididymis (*Figure 2—figure supplement 2D*). Cross-sections of testis seminiferous tubules from 8-week-old males revealed major degeneration of germ cells in *Ythdc2^-/-^* tubules compared to wild-type controls (*Figure 2D,E*). Examination of histological sections from P30 testes at high magnification revealed that *Ythdc2^-/-^* tubules contained spermatogonia as well as preleptotene spermatocytes with nuclear morphology similar to wild type (*Figure 2F,F′ and and G,G′′*, black arrowheads). However, normal appearing preleptotene cells in *Ythdc2^-/-^* tubules frequently had cells nearby that either had highly condensed chromosomes (*Figure 2G,G′*, red arrowhead and *Figure 2—figure supplement 2E*) or appeared apoptotic (*Figure 2G,G′′′*, red arrow and *Figure 2—figure supplement 2E*). In contrast, wild-type tubules rarely contained cells with condensed chromosomes or apoptotic cells near preleptotene cells (*Figure 2—figure supplement 2E*). Both metaphase and anaphase figures were detected near preleptotene spermatocytes in *Ythdc2^-/-^* tubules (*Figure 2H–H′′* and *Figure 2—figure supplement 2E*), suggesting that the abnormal metaphase cells had attempted to divide. Although not as frequently as in age-matched wild-type tubules, some P30 *Ythdc2^-/-^* tubules contained germ cells entering the

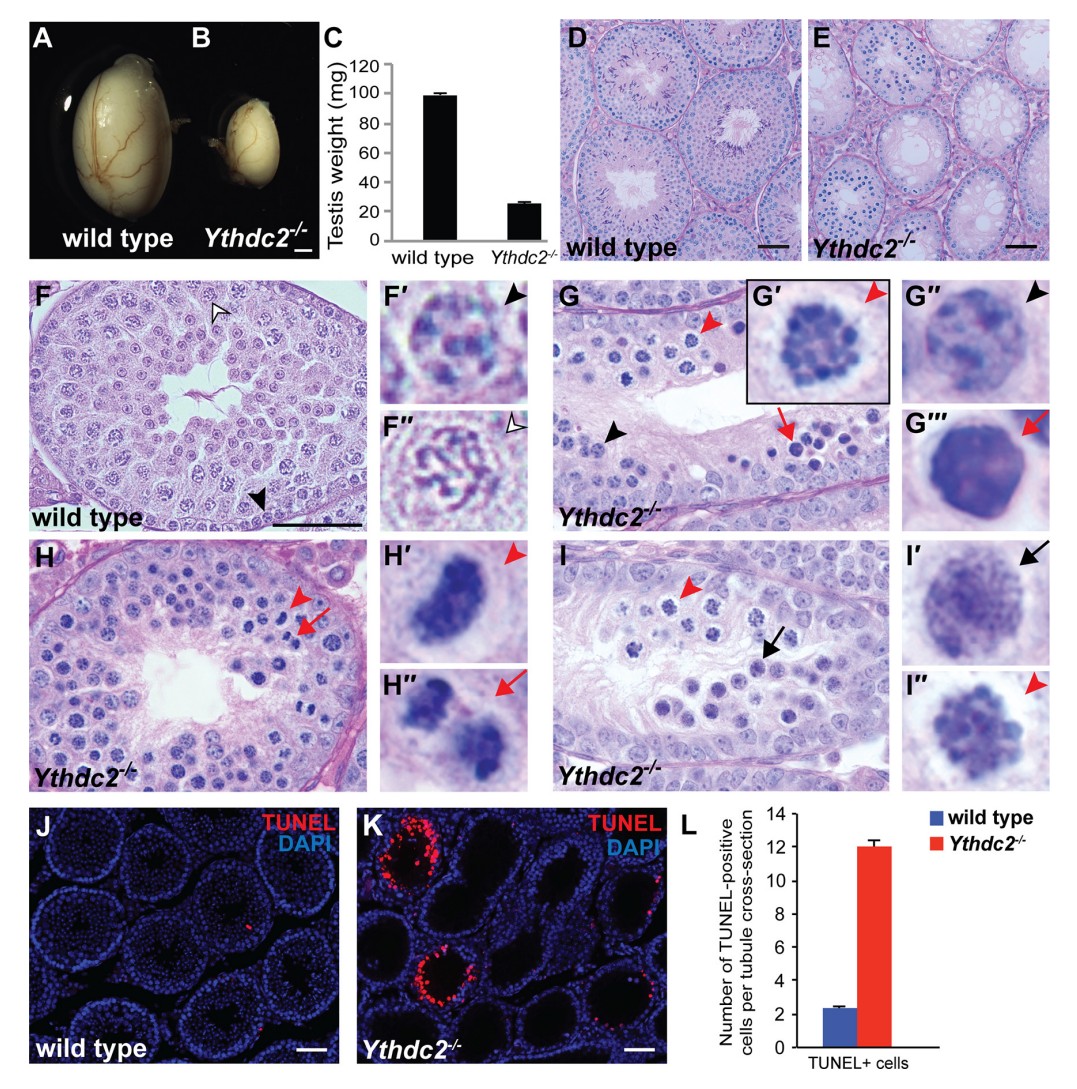

**Figure 2.** Adult *Ythdc2* mutant male germ cells fail to properly execute meiosis. (A, B) Whole mount images of (A) wild-type and (B) *Ythdc2⁻/⁻* testes from 12-week-old male mice. (C) Testis weights from 8-week-old adult wild-type and *Ythdc2⁻/⁻* mice (n = 6 testes per group; *t* test, p<0.0001). Error bars: SEM. (D–I) Sections of mouse seminiferous tubules stained with periodic acid-Schiff (PAS). (D, E) Testis tubules from 8-week-old (D) wild-type and (E) *Ythdc2⁻/⁻* mice (n = 3 mice per group). (F) Testis tubule cross-sections from a P30 wild-type mouse. Black arrowhead: preleptotene cell. Outlined arrowhead: pachytene spermatocyte. (F′, F″) High-magnification images of germ cells indicated in panel F. (G) Testis tubule from a P30 *Ythdc2⁻/⁻* mouse. Black arrowhead: normal appearing preleptotene cell. Red arrowhead: cell with mitotic-like condensed chromosomes. Red arrow: apoptotic cell. (G′-G‴) High magnification images of germ cells indicated in panel G. (H) Testis tubule from a P30 *Ythdc2⁻/⁻* mouse. Red arrowhead: germ cell in abnormal metaphase. Red arrow: germ cell in abnormal anaphase. (H′, H″) High-magnification images of germ cells indicated in panel H. (I) Testis tubule from a P30 *Ythdc2⁻/⁻* mouse. Black arrow: leptotene-like germ cell. Red arrowhead: germ cell with condensed chromosomes. (I′, I″) High-magnification images of germ cells indicated in panel I. n = 3 P30 mice per group. (J, K) Testes from P30 (J) wild-type and (K) *Ythdc2⁻/⁻* mice labeled with TUNEL (red) and DAPI (blue). (L) Number of TUNEL-positive cells per tubule cross-section in P30 wild-type (blue) and *Ythdc2⁻/⁻* (red) testes (n = 3 mice per group, two sections per mouse; *t* test, p<0.0001). Error bars: SEM. Scales bars: (A, B) 1 mm; (D–F and J, K) 50 μm. See also *Figure 2—figure supplement 1*, *Figure 2—figure supplement 2*, *Figure 2—source data 1* and *Figure 2—source data 2*.

DOI: https://doi.org/10.7554/eLife.26116.004

The following source data and figure supplements are available for figure 2:

**Source data 1.** Testis weights in adult wild-type and *Ythdc2⁻/⁻* mice.
DOI: https://doi.org/10.7554/eLife.26116.007

**Source data 2.** Number of TUNEL + cells per tubule cross-section in wild-type and *Ythdc2⁻/⁻* testes.
DOI: https://doi.org/10.7554/eLife.26116.008

**Source data 3.** Percentage of P30 tubule cross-sections containing indicated cell types in wild-type and *Ythdc2⁻/⁻* testes.

*Figure 2 continued on next page*

*Figure 2 continued*

DOI: https://doi.org/10.7554/eLife.26116.009

**Figure supplement 1.** Generation of the *Ythdc2* knockout mouse.

DOI: https://doi.org/10.7554/eLife.26116.005

**Figure supplement 2.** Phenotypic characterization of *Ythdc2* mutant mice.

DOI: https://doi.org/10.7554/eLife.26116.006

leptotene stage of meiotic prophase (*Figure 2I,I'*, black arrow and *Figure 2—figure supplement 2E*). However, *Ythdc2⁻/⁻* tubules lacked late spermatocyte and spermatid stages seen in wild-type tubules (*Figure 2F,F''*, outlined arrowhead). The transition from the preleptotene or leptotene stage, to abnormal chromosome condensation, to cell death appeared to occur rapidly, as all three stages were often observed within the same seminiferous tubule section in *Ythdc2⁻/⁻* testes (*Figure 2G*). TUNEL staining of P30 testes showed an increased number of TUNEL-positive cells in *Ythdc2⁻/⁻* tubules compared to wild type (*Figure 2J–L*). Apoptosis was widespread throughout P30 testes; however, due to differences in developmental stage, only a subset of *Ythdc2⁻/⁻* tubules in a given cross-section contained TUNEL-positive cells.

Consistent with the findings from adult testes, histologic analysis of the first wave of spermatogenesis showed defects in germ cell differentiation in *Ythdc2⁻/⁻* testes from early in meiotic prophase (*Figure 3A–H*). At P8, the time point when most germ cells are at the spermatogonial stage, *Ythdc2⁻/⁻* testis tubules were filled with spermatogonia that appeared normal compared to wild-type littermate controls (*Figure 3A,B*). However, germ cells in *Ythdc2⁻/⁻* testes began to show abnormalities at P10, when the first wave of germ cells normally initiate meiotic prophase. Wild-type tubules at P10 contained preleptotene as well as early leptotene (*Figure 3C*, black arrowheads and inset) spermatocytes. In contrast, although *Ythdc2⁻/⁻* tubules at P10 contained histologically normal preleptotene spermatocytes (*Figure 3D*, black arrows and inset), many tubules from P10 *Ythdc2⁻/⁻* mice had germ cells that started to condense their chromosomes, as if trying to divide rather than entering the normally extended meiotic prophase period (*Figure 3D*, red arrowheads). At P12, wild-type tubules contained many leptotene (*Figure 3E*, black arrowheads and inset) and zygotene spermatocytes. While some P12 *Ythdc2⁻/⁻* tubules contained leptotene spermatocytes (*Figure 3F*, black arrowheads and inset) and a few contained zygotene spermatocytes (*Figure 3I*), many tubules contained germ cells with condensed chromosomes (*Figure 3J*, red arrowheads and inset, and 3K) rarely seen in age-matched wild-type testes. By P14, many germ cells had entered the pachytene stage of meiotic prophase in wild-type testes (*Figure 3G*, outlined arrowheads and inset, and 3L-L''). However, no pachytene spermatocytes were observed in P14 *Ythdc2⁻/⁻* testis tubules and many tubules still contained germ cells with condensed chromosomes (*Figure 3H*, red arrowheads and inset). Analysis of P14 chromosome spreads, in which abnormal *Ythdc2⁻/⁻* germ cells with condensed chromosomes were distinguished from mitotic nuclei based on their expression of the meiotic marker SYCP3, revealed that 95% of *Ythdc2⁻/⁻* germ cells with condensed chromosomes had 40 chromosomes (*Figure 3M–M''* and 3N), similar to a mitotic division, rather than 20 paired bivalents typical for meiosis I. In addition, SYCP3 appeared to localize to foci at the ends of the condensed chromosomes in *Ythdc2⁻/⁻* germ cells (*Figure 3M–M''*) rather than along the chromosomes as in wild type (*Figure 3L–L''*).

Analysis of 5-ethynyl-2′-deoxyuridine (EdU) incorporation and staining for expression of STRA8, a key regulator of meiotic entry, confirmed that *Ythdc2⁻/⁻* germ cells reached the preleptotene stage and went through premeiotic S-phase. The expression pattern of STRA8 protein in *Ythdc2⁻/⁻* testes was similar to wild type (*Figure 4A,B*). After a short pulse of EdU followed by a 2 hr chase, many germ cells were positive for both STRA8 and EdU, indicating that preleptotene spermatocytes in *Ythdc2⁻/⁻* testes were successfully undergoing premeiotic DNA synthesis, similar to wild type (*Figure 4C,D*).

During the first wave of spermatogenesis, wild-type germ cells up-regulate several factors important for meiosis around P10-P12. Consistent with our histologic analysis, YTHDC2-deficient male germ cells expressed several early meiotic markers. Both wild-type and YTHDC2-deficient germ cells stained positive for the meiotic cohesion REC8 (*Figure 4E*), an early meiotic marker expressed starting in the preleptotene stage. Both wild-type and *Ythdc2⁻/⁻* P14 spreads also showed nuclei with SYCP3 expression (*Figure 4F*), confirming that some *Ythdc2* mutant germ cells enter the early stages

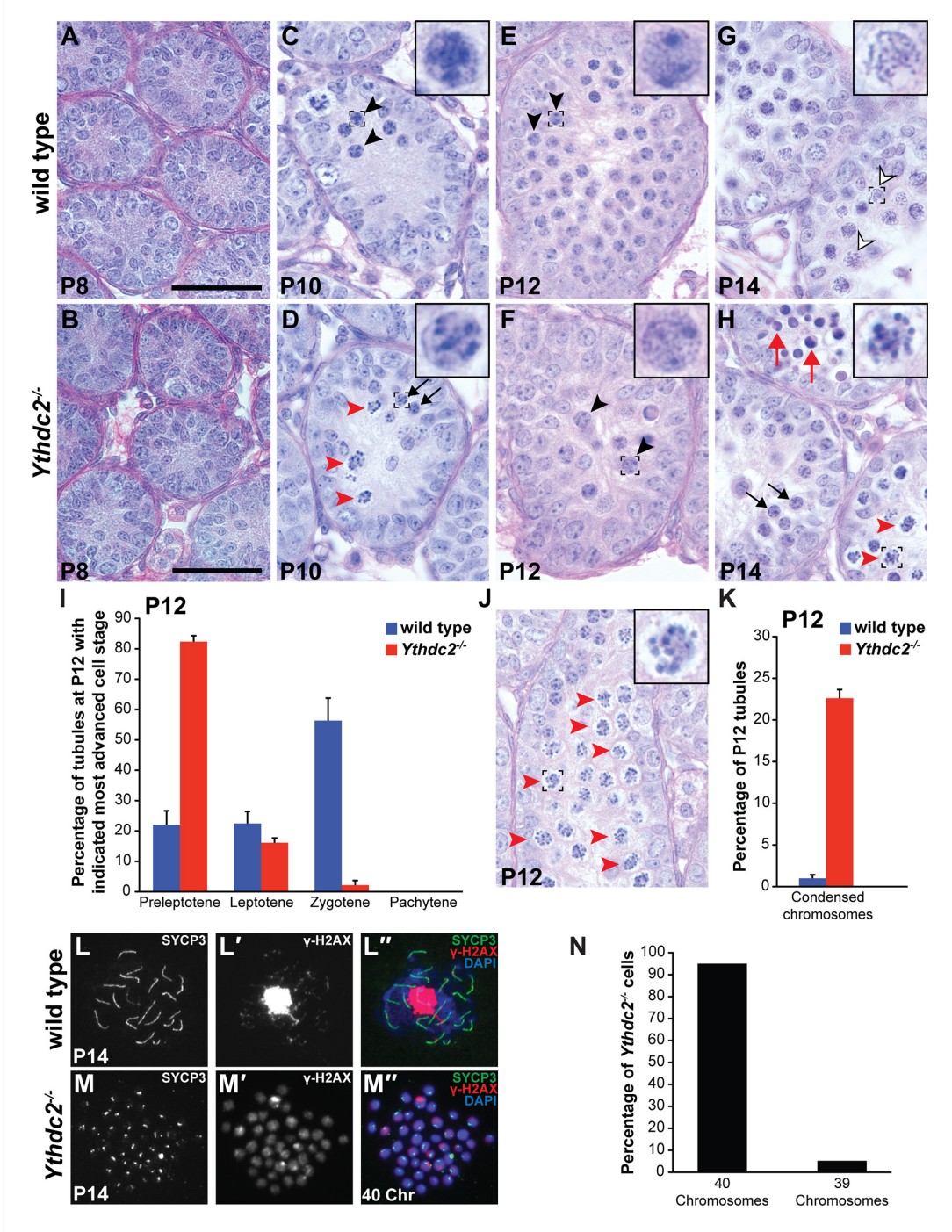

**Figure 3.** YTHDC2-deficient male germ cells do not properly progress through meiotic prophase during the first wave of spermatogenesis. (A–H) Testis seminiferous tubule cross-sections counterstained with PAS from wild-type (top) and *Ythdc2⁻/⁻* (bottom) mice during the first wave of spermatogenesis: (A, B) P8; (C, D) P10; (E, F) P12; and (G, H) P14 (n = 3 wild-type and *Ythdc2⁻/⁻* mice per time point). Black arrows: normal preleptotene cells. Black arrowheads: normal leptotene spermatocytes. Outlined arrowheads: normal pachytene spermatocytes. Red arrowheads: abnormal cells with mitotic-like condensed chromosomes. Red arrows: apoptotic cells. Insets are high-magnification images of representative boxed germ cells. Scale bars: 50 μm. (I) Percentage of P12 tubule cross-sections containing preleptotene, leptotene, zygotene or pachytene cells as the most advanced cell stage in wild-type (blue) and *Ythdc2⁻/⁻* (red) testes (n = 3 mice per group, ≥125 tubules counted per mouse). Error bars: SEM. (J) P12 *Ythdc2* mutant tubule containing many abnormal cells with condensed chromosomes (red arrowheads). (K) Percentage of P12 tubule cross-sections containing cells with condensed chromosomes in wild-type (blue) and *Ythdc2⁻/⁻* (red) testes (n = 3 mice per group, ≥125 tubules counted per mouse). Error bars: SEM. (L, M) Immunofluorescence images of germ cell spreads from P14 (L-L'') wild-type and (M-M'') *Ythdc2⁻/⁻* mice stained for SYCP3 (green), γ-H2AX (red) and

*Figure 3 continued on next page*

*Figure 3 continued*

DAPI (blue). (**N**) Percentage of *Ythdc2*$^{-/-}$ metaphase-like nuclei that contain 40 chromosomes (counted nuclei from n = 4 *Ythdc2*$^{-/-}$ mice; 77 *Ythdc2*$^{-/-}$ nuclei counted). See also *Figure 3—source data 1*, *Figure 3—source data 2* and *Figure 3—source data 3*.

DOI: https://doi.org/10.7554/eLife.26116.010

The following source data is available for figure 3:

**Source data 1.** The most advanced cell stage in P12 wild-type and *Ythdc2*$^{-/-}$ testis tubules.
DOI: https://doi.org/10.7554/eLife.26116.011

**Source data 2.** Percentage of P12 wild-type and *Ythdc2*$^{-/-}$ testis tubules containing cells with condensed chromosomes.
DOI: https://doi.org/10.7554/eLife.26116.012

**Source data 3.** Number of chromosomes in *Ythdc2*$^{-/-}$ metaphase-like nuclei.
DOI: https://doi.org/10.7554/eLife.26116.013

of meiotic prophase. Analysis of germ cell spreads confirmed that YTHDC2-deficient germ cells can reach leptotene (*Figure 4F* and *Figure 4—figure supplement 1A*) and early zygotene (*Figure 4—figure supplement 1B*) spermatocyte stages. However, despite entering the early stages of meiosis, YTHDC2-deficient male germ cells failed to properly express several meiotic markers. In contrast to the high levels of DMC1 protein in wild-type spermatocytes at P12, high levels of DMC1 were detected by immunofluorescence in only a small percentage of *Ythdc2*$^{-/-}$ tubules (*Figure 4G–I* and *Figure 4—figure supplement 1C,D*). Analysis of expression of *Spo11* mRNA in testis extracts showed a decrease in *Ythdc2* mutant compared to wild-type testes at P14 (*Figure 4J*). By P14, most tubule cross-sections from wild type had high levels of SYCP3 that localized along the entire length of the chromosomes (*Figure 4K,L*). In contrast, while SYCP3 was observed in some *Ythdc2*$^{-/-}$ testes tubule cross-sections at P14 (*Figure 4M*), the level of SYCP3 protein detected was usually low and often localized in a punctate pattern, as in earlier stages of meiosis (*Figure 4N*).

## YTHDC2 is required for meiotic prophase in the fetal ovary

*Ythdc2*$^{-/-}$ adult female mice were infertile and had thin uteri and small ovaries compared to wild type (*Figure 5A,B*). Ovaries from adult *Ythdc2*$^{-/-}$ mice completely lacked developing follicles (*Figure 5—figure supplement 1B*). While P21 wild-type ovaries contained full-grown oocytes in germinal vesicles (*Figure 5C*), no germinal vesicles were detected in P21 *Ythdc2*$^{-/-}$ ovaries (*Figure 5D*). At P5, when wild-type ovaries were filled with primordial follicles, no primordial follicles were detected in *Ythdc2*$^{-/-}$ ovaries (*Figure 5—figure supplement 1C,D*). At birth (P0.5), *Ythdc2*$^{-/-}$ ovaries contained some DDX4$^+$ germ cells, but many fewer than in aged-matched wild-type ovaries (*Figure 5—figure supplement 1E–G*), suggesting that YTHDC2 function is required during the fetal period for survival of female germ cells.

Histologic analysis of fetal ovaries revealed defects in *Ythdc2* mutants starting between E14.5 and E15.5, the time when female germ cells normally enter meiotic prophase. At E14.5, *Ythdc2*$^{-/-}$ ovaries contained many premeiotic germ cells that appeared similar to those in wild type (*Figure 5E,F*). By E15.5, however, when many wild-type germ cells had entered meiotic prophase (*Figure 5G*), YTHDC2-deficient germ cells did not show the chromosomal changes characteristic of meiotic prophase, but instead showed nuclear morphology resembling premeiotic cells (*Figure 5H*). At E16.5, when wild-type germ cells were clearly progressing through meiotic prophase (*Figure 5I*), *Ythdc2*$^{-/-}$ germ cells showed abnormal chromatin condensation (*Figure 5J*). Consistent with the appearance of defects in germ cells in E15.5 *Ythdc2*$^{-/-}$ ovaries, YTHDC2 protein was up-regulated at E15.5 in wild type (*Figure 5—figure supplement 1H*) in germ cells that co-expressed the meiotic marker SYCP3 (*Figure 5—figure supplement 1I*).

The apparent defects in the transition from mitosis to meiosis seen in *Ythdc2*$^{-/-}$ ovaries were not due to low levels of STRA8, as immunofluorescence staining of E14.5 *Ythdc2*$^{-/-}$ ovaries revealed similar levels and distribution of STRA8 protein as in age-matched wild-type ovaries (*Figure 5K,L*). Some germ cells in *Ythdc2*$^{-/-}$ ovaries expressed SYCP3 at E15.5 (*Figure 5—figure supplement 1J,K*), suggesting that, as in the male, some YTHDC2-deficient female germ cells enter the early stages of meiosis. Also similar to males, few YTHDC2-deficient female germ cells expressed DMC1 at E15.5 (*Figure 5N*), in contrast to the abundant expression of DMC1 in age-matched wild-type ovaries (*Figure 5M*). Consistent with failure of *Ythdc2*$^{-/-}$ female germ cells to properly transition from mitotic

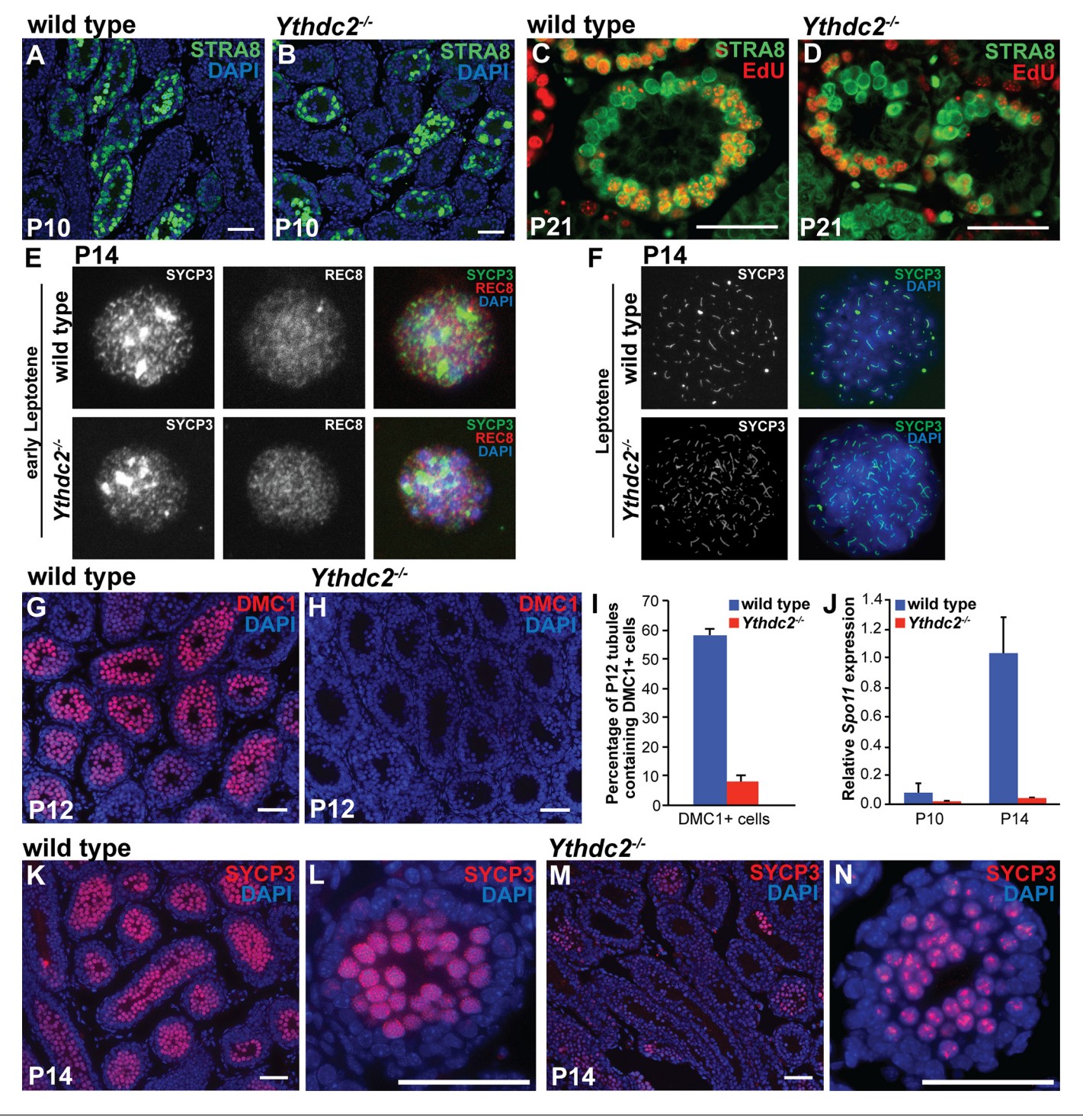

**Figure 4.** Defects in expression of some meiotic markers in *Ythdc2* mutant male germ cells. (A–D) Immunofluorescence images of testis tubule cross-sections. (A, B) Testes from P10 (A) wild-type and (B) *Ythdc2*−/− mice stained for STRA8 (green) and DAPI (blue) (n = 3 mice per group). (C, D) Testes from P21 (C) wild-type and (D) *Ythdc2*−/− mice labeled with EdU (red) and stained for STRA8 (green) (n = 3 mice per group). (E) Immunofluorescence images of early leptotene germ cells from P14 wild-type and *Ythdc2*−/− mice stained for SYCP3 (green), REC8 (red) and DAPI (blue). (F) Immunofluorescence images of leptotene germ cells from P14 wild-type and *Ythdc2*−/− mice stained for SYCP3 (green) and DAPI (blue). (G, H) Testes from P12 (G) wild-type and (H) *Ythdc2*−/− mice stained for DMC1 (red) and DAPI (blue). (I) Percentage of P12 testis tubules containing DMC1+ germ cells in wild-type (blue) and *Ythdc2*−/− (red) mice (n = 4 mice per group, one cross-section per mouse with >115 tubules per cross-section; t test, p<0.0001). Error bars: SEM. (J) Quantitative RT-PCR analysis of *Spo11* mRNA expression in testes from P10 and P14 wild-type (blue) and *Ythdc2*−/− (red) mice (n = 2

*Figure 4 continued on next page*

*Figure 4 continued*

mice per group). Error bars: SEM. (**K–N**) Testes from P14 (**K, L**) wild-type and (**M, N**) *Ythdc2⁻/⁻* mice stained for SYCP3 (red) and DAPI (blue). (**L, N**) High magnification images of single P14 testis tubules from (**L**) wild-type and (**N**) *Ythdc2⁻/⁻* mice. Scale bars: 50 µm. See also ***Figure 4—figure supplement 1***, ***Figure 4—source data 1*** and ***Figure 4—source data 2***.

DOI: https://doi.org/10.7554/eLife.26116.014

The following source data and figure supplement are available for figure 4:

**Source data 1.** Percentage of tubules containing DMC1 + germ cells in wild-type and *Ythdc2⁻/⁻* P12 testes.

DOI: https://doi.org/10.7554/eLife.26116.016

**Source data 2.** Expression of *Spo11* mRNA in wild-type and *Ythdc2⁻/⁻* testes.

DOI: https://doi.org/10.7554/eLife.26116.017

**Figure supplement 1.** Expression of meiotic markers in *Ythdc2* mutant males.

DOI: https://doi.org/10.7554/eLife.26116.015

divisions to meiotic prophase, expression of the G1 mitotic marker Cyclin D1 appeared to persist in many germ cells in *Ythdc2⁻/⁻* ovaries at E15.5 (***Figure 5P***), while very few germ cells expressed Cyclin D1 in wild-type ovaries at this stage (***Figure 5O***). By E16.5, wild-type germ cells had mostly ceased mitosis and entered meiotic prophase (***Figure 5Q***). However, consistent with the abnormal condensation of chromosomes observed in histological sections (***Figure 5J***), 16% of DDX4⁺ germ cells stained positive for pH3 in E16.5 *Ythdc2⁻/⁻* ovaries (***Figure 5R*** and ***Figure 5—figure supplement 1L***) compared to only 1.4% in wild-type ovaries at E16.5 (***Figure 5Q*** and ***Figure 5—figure supplement 1L***), suggesting that in the absence of YTHDC2 function, female germ cells failed to properly implement meiotic prophase in the fetal ovary.

The *Ythdc2* mutant phenotype is similar to the *Meioc* mutant phenotype in both males and females (***Abby et al., 2016***; ***Soh et al., 2017***). Consistent with this, immunoprecipitation from testis extracts showed that YTHDC2 physically interacts with MEIOC (***Figure 5—figure supplement 2A***), suggesting that the two proteins may work together to promote progression through meiotic prophase. *Meioc* RNA expression was not significantly different in P12 *Ythdc2⁻/⁻* testes compared to wild type (Figure 8C and ***Figure 6—source data 1***). However, MEIOC protein levels were much lower in *Ythdc2⁻/⁻* P12 testes than in wild type, being barely detected by western blot (***Figure 5—figure supplement 2B***) or by immunofluorescence staining (***Figure 5—figure supplement 2D***) in the mutant.

## *Ythdc2⁻/⁻* male germ cells fail to turn off expression of Cyclin A2 and properly maintain the spermatocyte transcription program

Comparison of the mRNAs expressed in *Ythdc2⁻/⁻* and age-matched wild-type testes at P12 (***Figure 6A***) and P14 (***Figure 6B***) by RNA sequencing revealed a large number of transcripts (959 transcripts at P14, adjusted p-value<0.05) that were expressed at lower levels in *Ythdc2⁻/⁻* testes than in age-matched wild-type testes. While the changes at P12 were subtle, the trend was similar: 199 transcripts were expressed at lower levels in *Ythdc2⁻/⁻* than in wild-type testes. The transcripts most under-expressed in *Ythdc2⁻/⁻* compared to age-matched wild-type testes were frequently up-regulated in spermatocytes compared to spermatogonia based on data from sorted germ cells (***Soumillon et al., 2013***) (***Figure 6—source data 1*** and ***Figure 6—source data 2***). Seventy-three percent of the transcripts under-expressed in P12 *Ythdc2⁻/⁻* versus wild-type testes were up-regulated in wild-type spermatocytes compared to spermatogonia (p<0.0001, chi-square). Eighty-six percent of the transcripts under-expressed in P14 *Ythdc2* mutant testes were up-regulated in wild-type spermatocytes compared to spermatogonia (p<0.0001, chi-square). Together, these observations suggest widespread failure to express or maintain the spermatocyte transcription program in *Ythdc2⁻/⁻* male germ cells. The RNA-Seq data are consistent with our finding that the level of *Spo11* mRNA was decreased in P14 *Ythdc2⁻/⁻* testes compared to wild type by qRT-PCR (***Figure 4J***). In fact, at the transcriptome level, P14 *Ythdc2⁻/⁻* cells were more similar to P12 wild-type cells than to P14 wild-type cells (***Figure 6—figure supplement 1***). Consistent with this, GO-term analysis revealed that for both P12 and P14, the mRNAs most under-expressed in *Ythdc2⁻/⁻* versus wild-type age-matched controls were enriched for mRNAs associated with male meiosis and the meiotic cell cycle (***Figure 6C***).

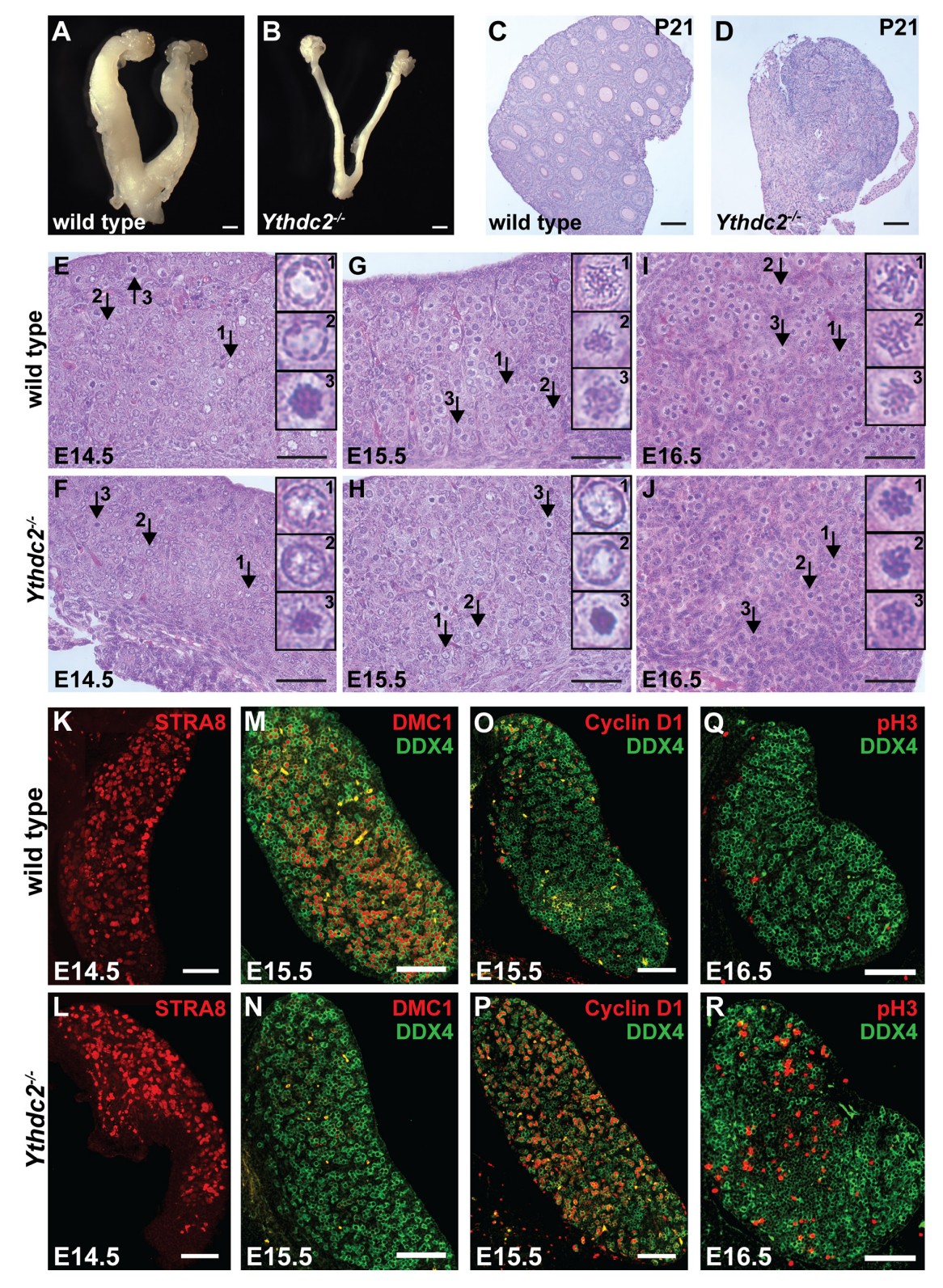

**Figure 5.** YTHDC2 is required for meiotic prophase in the fetal ovary. (A, B) Whole mount ovaries from 8-week-old adult female (A) wild-type and (B) *Ythdc2*<sup>-/-</sup> mice (n = 3 wild-type and n = 5 *Ythdc2*<sup>-/-</sup> mice). (C, D) PAS-stained ovary sections from P21 (C) wild type (8/8 ovaries contained germinal vesicles) and (D) *Ythdc2*<sup>-/-</sup> (0/7 ovaries contained germinal vesicles). (E–J) Embryonic ovary cross-sections counterstained with H and E from wild-type and *Ythdc2*<sup>-/-</sup> mice: (E, F) E14.5 (n = 3 wild-type and n = 3 *Ythdc2*<sup>-/-</sup> embryos); (G, H) E15.5 (n = 4 wild-type and n = 4 *Ythdc2*<sup>-/-</sup> embryos); and (I, J) E16.5

*Figure 5 continued on next page*

*Figure 5 continued*

(n = 3 wild-type and n = 5 *Ythdc2*^-/- embryos). Arrows: representative germ cells shown in insets. (**K–R**) Immunofluorescence images of fetal ovary cross-sections. (**K, L**) E14.5 ovaries from (**K**) wild-type and (**L**) *Ythdc2*^-/- mice stained for STRA8. (**M, N**) E15.5 ovaries from (**M**) wild-type and (**N**) *Ythdc2*^-/- mice stained for DMC1 (red) and DDX4 (green). (**O, P**) E15.5 ovaries from (**O**) wild-type and (**P**) *Ythdc2*^-/- mice stained for Cyclin D1 (red) and DDX4 (green). (**Q, R**) E16.5 ovaries from (**Q**) wild-type and (**R**) *Ythdc2*^-/- mice stained for pH3 (red) and DDX4 (green). Scale bars: (**A, B**) 1 mm; (**C, D**) 100 μm; (**E–J**) 50 μm; (**K–R**) 100 μm. See also ***Figure 5—figure supplement 1*** and ***Figure 5—figure supplement 2***.
DOI: https://doi.org/10.7554/eLife.26116.018

The following source data and figure supplements are available for figure 5:

**Source data 1.** Number of DDX4 + cells in wild-type and *Ythdc2*^-/- P0.5 ovaries.
DOI: https://doi.org/10.7554/eLife.26116.021

**Source data 2.** Percentage of DDX4 + cells that are pH3 + in wild-type and *Ythdc2*^-/- E16.5 ovaries.
DOI: https://doi.org/10.7554/eLife.26116.022

**Figure supplement 1.** *Ythdc2* mutant females have defects in the fetal ovary.
DOI: https://doi.org/10.7554/eLife.26116.019

**Figure supplement 2.** MEIOC coimmunoprecipitates with YTHDC2.
DOI: https://doi.org/10.7554/eLife.26116.020

In contrast, 19 transcripts remained subtly elevated in P12 *Ythdc2*^-/- testes compared to age-matched wild-type controls. Based on published datasets, these were encoded by genes that tend to be expressed in spermatogonia but normally down-regulated in wild-type spermatocytes (***Figure 6—source data 1***), raising the possibility that the mitotic program may fail to properly shut down in the absence of YTHDC2 function. A striking example is Cyclin A2, which plays a role in both S phase and G2/M of the cell cycle (***Satyanarayana and Kaldis, 2009***). Cyclin A2 is highly expressed in spermatogonia undergoing mitotic divisions along the periphery of wild-type testis tubules (***Figure 7A–A''***, arrow) (***Wolgemuth et al., 2013***). Cyclin A2 expression appears to decrease substantially in wild-type preleptotene cells (***Figure 7—figure supplement 1A–A''***, arrowheads) and is normally fully down-regulated in wild-type germ cells by entry into the leptotene stage at both the protein (***Figure 7A–A''***, arrowheads and 7C) and RNA level (***Ravnik and Wolgemuth, 1996***). Immunofluorescence analysis revealed that, in contrast to wild type, Cyclin A2 protein levels were high in YTHDC2-deficient preleptotene spermatocytes that moved inward from the periphery of the testis tubule (***Figure 7—figure supplement 1B–B''***, arrowheads), and remained high in YTHDC2-deficient germ cells that exhibited SYCP3 staining typical of leptotene spermatocytes (***Figure 7B–B''***, arrowheads and ***Figure 7D,E***).

## YTHDC2 binds specific RNA targets and localizes to RNA granules

Consistent with its multiple predicted RNA-binding domains, YTHDC2 appears to bind a number of RNAs in testis extracts. Light formaldehyde fixation followed by immunoprecipitation of YTHDC2 (fRIP) and sequencing of co-immunoprecipitated RNAs from P12 wild-type versus *Ythdc2* mutant testes revealed 973 RNAs enriched >2 fold (adjusted p-value<0.05) in the wild-type compared to *Ythdc2*^-/- samples (***Figure 8—source data 1***). The set of mRNAs identified as bound by YTHDC2 in the fRIP studies was enriched for transcripts involved in the cell cycle program and mitotic prophase (***Figure 8A***). Among these bound mRNAs was *Ccna2*, which encodes Cyclin A2. Enrichment of *Ccna2* by immunoprecipitation of YTHDC2 was confirmed by independent fRIP followed by qRT-PCR (***Figure 8B***). fRIP-Seq for YTHDC2 enriched *Ccna2* mRNA 2.4-fold (p-value=$4.7E^{-13}$), suggesting that YTHDC2 may normally act directly to down-regulate expression of *Ccna2* in early meiotic prophase. As the level of *Ccna2* mRNA was slightly higher in P12 *Ythdc2* mutant testes compared to wild type (***Figure 8C***), action of YTHDC2 to down-regulate Cyclin A2 expression in wild-type spermatocytes could be through either RNA stability or via translational control.

In addition to *Ccna2*, YTHDC2 bound a number of other transcripts that function during the mitotic cell cycle, including *Rad21* and *Wee1*, suggesting that YTHDC2 may act on a number of mitotic transcripts as germ cells enter meiotic prophase. The RNA levels of *Rad21* and *Wee1* were similar in P12 wild-type and *Ythdc2* mutant testes. In fact, for the majority of the 973 RNAs enriched >2 fold by fRIP for YTHDC2 from wild-type P12 testes, the level of RNA expression changes was under two-fold between between wild-type and *Ythdc2* mutant testes at both P12 and P14 (***Figure 8—figure supplement 1***). Based on the fRIP-Seq data, YTHDC2 also appeared to bind

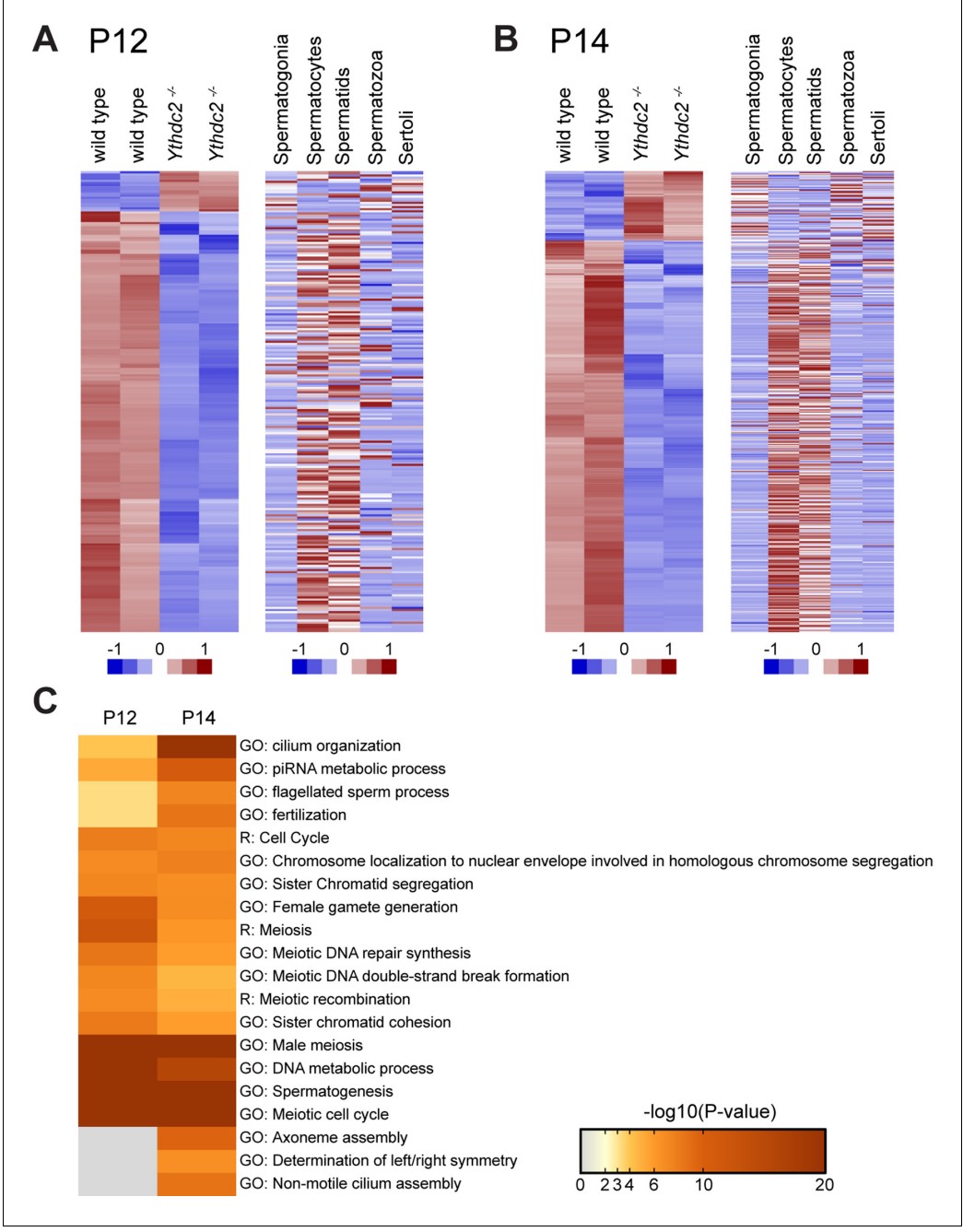

**Figure 6.** *Ythdc2⁻/⁻* male germ cells do not properly up-regulate meiotic transcripts at the mitosis to meiosis transition. (**A, B**) Heatmaps representing transcripts differentially expressed in *Ythdc2⁻/⁻* compared to wild type in (**A**) P12 and (**B**) P14 testes (adjusted p-value<0.05), with corresponding expression levels for the same genes in sorted cell populations from adult testes (***Soumillon et al., 2013***). Biological replicates for both wild type and *Ythdc2⁻/⁻* are shown. For P12: 218 total transcripts shown, 199 transcripts down in *Ythdc2⁻/⁻* compared to wild type and 19 transcripts higher in *Ythdc2⁻/⁻* compared to wild type. For P14: 1130 total transcripts shown, 959 transcripts down in *Ythdc2⁻/⁻* compared to wild type and 171 transcripts higher in *Ythdc2⁻/⁻* compared to wild type. (**C**) Heatmap representing enrichment of GO and Reactome (R) -terms for the genes expressed at lower levels in P12 and P14 *Ythdc2⁻/⁻* testes compared to age-matched wild-type testes. See also ***Figure 6—figure supplement 1***, ***Figure 6—source data 1*** for the list of differentially expressed genes in P12 *Ythdc2* mutant testes compared to wild type and ***Figure 6—source data 2*** for the list of differentially expressed genes in P14 *Ythdc2* mutant testes compared to wild type.

DOI: https://doi.org/10.7554/eLife.26116.023

*Figure 6 continued on next page*

*Figure 6 continued*

The following source data and figure supplement are available for figure 6:

**Source data 1.** Differentially expressed transcripts in P12 *Ythdc2⁻/⁻* testes compared to wild type.
DOI: https://doi.org/10.7554/eLife.26116.025

**Source data 2.** Differentially expressed transcripts in P14 *Ythdc2⁻/⁻* testes compared to wild type.
DOI: https://doi.org/10.7554/eLife.26116.026

**Figure supplement 1.** P12 and P14 RNA-sequencing dataset analysis.
DOI: https://doi.org/10.7554/eLife.26116.024

*Meioc* mRNA (enriched 7.7-fold, p-value=$1.5E^{-17}$), which had a similar level of expression in wild-type and *Ythdc2⁻/⁻* P12 testes (*Figure 8C*). Independent fRIP-qRT-PCR confirmed that IP of YTHDC2 from P12 testes enriched for *Meioc* RNA (*Figure 8B*).

YTHDC2 contains a YTH domain capable of binding RNA with the N6-methyladenosine ($m^6A$) modification in the context of the RRACH motif (*Patil et al., 2016*; *Xu et al., 2015*). The $m^6A$ to A ratio in poly-A selected mRNAs from P12 *Ythdc2* mutant testes was similar to wild type (*Figure 8—figure supplement 2A*), suggesting that YTHDC2 activity is not required for formation or turnover of the $m^6A$ modification. Immunoprecipitation of $m^6A$ from poly-A selected RNA from P12 wild-type testes followed by high-throughput sequencing showed that $m^6A$ peaks were enriched around the stop codon (*Figure 8—figure supplement 2B*) and occurred at the RRACH motif as previously described (*Dominissini et al., 2012*; *Meyer et al., 2012*). In addition, the $m^6A$ IP revealed significant overlap between the mRNAs containing the $m^6A$-modification and the mRNAs enriched by fRIP for YTHDC2 in P12 testes. Indeed, 76% (735/969) of the mRNAs enriched by fRIP for YTHDC2 were also enriched by IP with anti-$m^6A$. In contrast, of the mRNAs not enriched by fRIP for YTHDC2, 47% (8,128/17,534) were enriched by IP with anti-$m^6A$. The difference (p<0.0001, chi-square) raises the possibility that the $m^6A$ modification may contribute to YTHDC2 target selectivity.

In addition to mRNAs, fRIP for YTHDC2 also enriched 7 piRNA precursor transcripts, including the pachytene piRNA precursor derived from the bidirectional piRNA cluster region 17-qA3.3–27363.1 (*Figure 8D*). Independent fRIP followed by qRT-PCR confirmed that IP of YTHDC2 from P12 testes enriched for the piRNA precursor over 20-fold (*Figure 8E*). RNA-Seq revealed that at P12 most of the piRNA precursors had a similar level of expression in wild-type and *Ythdc2* mutant testes, whereas piRNA precursor 17-qA3.3–27363.1 was present at much lower levels in P12 *Ythdc2⁻/⁻* testes compared to age-matched wild-type testes. Independent qRT-PCR confirmed the decrease in 17-qA3.3–27363.1 piRNA precursor transcript level in *Ythdc2⁻/⁻* testes (*Figure 8C*).

Pachytene piRNA precursor transcripts are enriched in germ granules (*Meikar et al., 2014*), which are key centers for post-transcriptional gene regulation in germ cells. Analysis of the subcellular localization of YTHDC2 in spermatocytes by immunofluorescence staining revealed YTHDC2 protein distributed throughout the cytoplasm but also concentrated in perinuclear puncta (*Figure 9A*). Co-staining with an antibody against the decapping enzyme DCP1A, a component of processing bodies (P-bodies), revealed that the cytoplasmic puncta containing YTHDC2 also contained DCP1A (*Figure 9A''*). Human YTHDC2 also concentrated in distinct perinuclear puncta in early spermatocytes, a stage marked by γ-H2AX foci (*Figure 9B*), suggesting that YTHDC2 may play a role in RNA granules in both mouse and human spermatocytes. In later spermatocytes, where γ-H2AX is only present in the sex vesicle, human YTHDC2 was detected at high levels throughout the cytoplasm (*Figure 9C*), similar to the high levels of cytoplasmic YTHDC2 in mouse pachytene spermatocytes (*Figure 1E*). Immunoprecipitation of YTHDC2 from wild-type adult mouse testes brought down the mouse PIWI family protein MIWI and the RNA binding protein MSY2 (*Figure 9D*). Reciprocal immunoprecipitation of either MSY2 (*Figure 9E*) or MIWI (*Figure 9F*) from adult mouse testes co-immunoprecipitated YTHDC2, suggesting that YTHDC2 may act together with MSY2 and MIWI, both components of RNA germ granules, to regulate translation or processing of target RNAs.

## Discussion

Our data suggest that the RNA-binding protein YTHDC2 promotes a clean transition from the mitotic to the meiotic program in mouse, reminiscent to the role of its homolog Bgcn in *Drosophila* males, indicating that deeply conserved molecular mechanisms based on post-transcriptional control

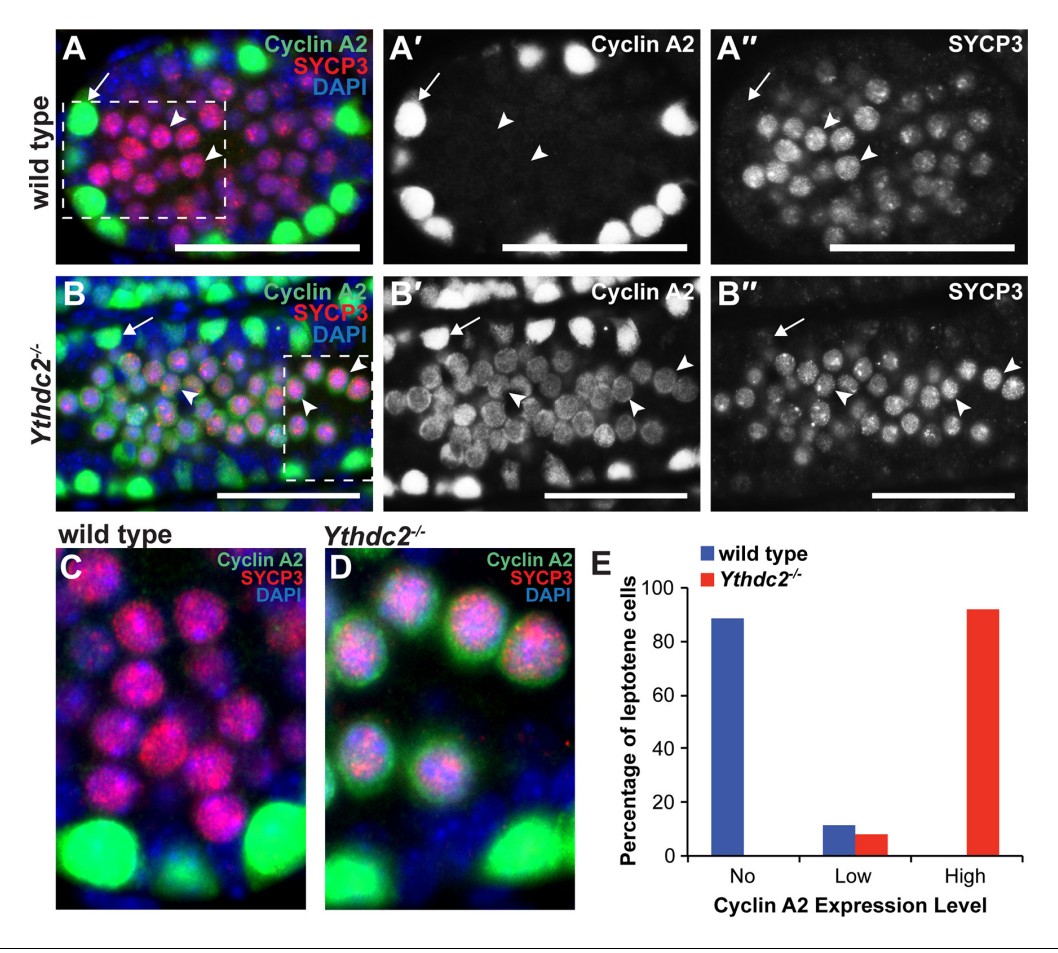

**Figure 7.** YTHDC2-deficient meiotic germ cells fail to turn off expression of Cyclin A2. (A-B″) Immunofluorescence images of testis cross-sections from P12 (A-A″) wild-type and (B-B″) *Ythdc2⁻/⁻* mice stained in parallel for Cyclin A2 (green), SYCP3 (red) and DAPI (blue). Arrow: spermatogonia. Arrowheads: leptotene spermatocytes. Scale bars: (A-B″) 50 µm. (C) High magnification of boxed region in panel A. (D) High magnification of boxed region in panel B. (E) Quantification of Cyclin A2 protein expression levels in leptotene spermatocytes in P12 wild-type (blue) or *Ythdc2* mutant (red) testes (counted leptotene spermatocytes from n = 3 wild-type and n = 3 *Ythdc2⁻/⁻* mice; >365 leptotene spermatocytes counted for both wild-type and *Ythdc2⁻/⁻* mice). See also *Figure 7—figure supplement 1* and *Figure 7—source data 1*.

DOI: https://doi.org/10.7554/eLife.26116.027

The following source data and figure supplement are available for figure 7:

**Source data 1.** Cyclin A2 protein expression in wild-type and *Ythdc2⁻/⁻* leptotene spermatocytes.
DOI: https://doi.org/10.7554/eLife.26116.029

**Figure supplement 1.** Cyclin A2 protein expression in wild-type and *Ythdc2* mutant preleptotene cells.
DOI: https://doi.org/10.7554/eLife.26116.028

of RNAs underlie the switch from mitosis to meiosis and terminal differentiation. We propose that action of YTHDC2 through post-transcriptional control of RNA is required during the early stages of meiotic entry in mice to both down-regulate the previous mitotic program and facilitate proper expression of meiotic and differentiation genes. Mouse YTHDC2-deficient male and female germ cells initiate entry into meiosis, as they express STRA8 (*Anderson et al., 2008*; *Baltus et al., 2006*) as well as early meiotic markers such as SYCP3 and REC8. However, YTHDC2-deficient male germ cells failed to accumulate high levels of several meiotic proteins and many meiotic transcripts and instead maintained abnormal expression of Cyclin A2 into the leptotene stage of meiotic prophase. The failure to adequately shut down the mitotic program manifested at the cytological level as well, as YTHDC2-deficient male germ cells never reached the pachytene stage, but instead condensed

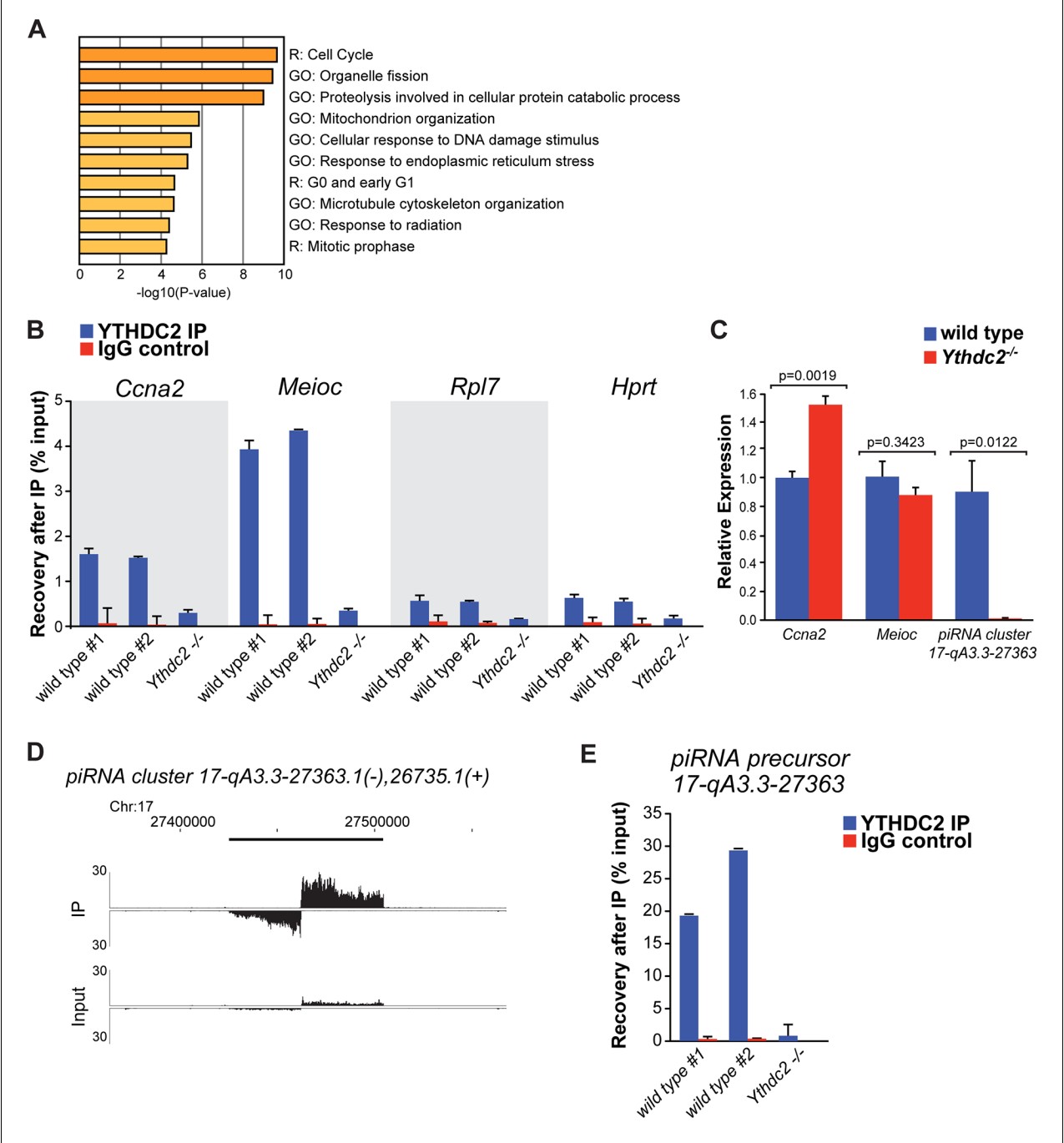

**Figure 8.** YTHDC2 binds specific RNA targets. (A) Top GO and Reactome (R) -terms for mRNAs enriched by immunoprecipitation of YTHDC2 from wild-type P12 testes. (B) RT-qPCR analysis of RNA transcripts following fRIP for YTHDC2 (blue) or IgG (red) from formaldehyde cross-linked P12 testes. (C) Quantitative RT-PCR analysis of the expression levels of YTHDC2 interacting RNA transcripts *Ccna2, Meioc*, and the piRNA precursor RNA (cluster region 17-qA3.3–27363.1) in testes from P12 wild-type (blue) and *Ythdc2-/-* (red) mice (n = 3 mice per group; *t* test, p-values indicated on plot). Error bars: SEM. (D) UCSC Genome Browser plots of read coverage for input and fRIP sequencing libraries. The Y-axis represents normalized coverage of reads. (E) RT-qPCR analysis of the piRNA precursor cluster region 17-qA3.3–27363.1 following fRIP for YTHDC2 (blue) or IgG (red) from formaldehyde cross-linked P12 testes. See also *Figure 8—figure supplement 1*, *Figure 8—figure supplement 2*, *Figure 8—source data 1* for the list of target RNAs identified by YTHDC2 immunoprecipitation after formaldehyde crosslinking, *Figure 8—source data 2* and *Figure 8—source data 3*.
DOI: https://doi.org/10.7554/eLife.26116.030

The following source data and figure supplements are available for figure 8:

**Source data 1.** RNAs enriched by fRIP for YTHDC2 from P12 testes.

*Figure 8 continued on next page*

*Figure 8 continued*

DOI: https://doi.org/10.7554/eLife.26116.033
**Source data 2.** Validation of YTHDC2-bound RNAs by fRIP-qRT-PCR from P12 testes.
DOI: https://doi.org/10.7554/eLife.26116.034
**Source data 3.** Analysis of expression levels of YTHDC2-bound RNAs by qRT-PCR in P12 wild-type and *Ythdc2*⁻/⁻ testes.
DOI: https://doi.org/10.7554/eLife.26116.035
**Figure supplement 1.** RNA expression levels of YTHDC2-bound RNAs in *Ythdc2* mutant testes relative to wild type.
DOI: https://doi.org/10.7554/eLife.26116.031
**Figure supplement 2.** The m⁶A to A ratio is similar in wild-type and *Ythdc2* mutant testes.
DOI: https://doi.org/10.7554/eLife.26116.032

their chromosomes early and entered a premature metaphase resembling mitosis, with 40 chromosomes rather than 20 bivalents. The co-expression of both mitotic and meiotic markers in YTHDC2-deficient male germ cells suggests that the cells have a mixed identity, which may contribute to the rapid loss of the mutant spermatocytes by apoptosis. Down-regulation of Cyclin A2 or other mitotic G2/M regulators in meiotic prophase may be critical to allow germ cells to set up the extended G2 period of meiotic prophase and prevent premature entry into M phase before homologous chromosome have properly paired, undergone recombination, and modified their kinetochores. In the mouse ovary, YTHDC2-deficient germ cells maintain expression of the G1 mitotic marker Cyclin D1 and also appear to enter metaphase rather than go into an extended meiotic prophase.

Results from immunoprecipitation of mouse YTHDC2 from wild-type P12 testis extracts followed by high-throughput sequencing (fRIP-Seq) revealed that YTHDC2 fRIP enriched several RNAs involved in the mitotic cell cycle, including *Ccna2*, *Rad21* and *Wee1*. Our finding that Cyclin A2 protein was expressed in *Ythdc2* mutant leptotene spermatocytes while absent in wild-type leptotene spermatocytes, suggests that YTHDC2 may be required to directly down-regulate expression of Cyclin A2 as germ cells enter meiotic prophase. fRIP for YTHDC2 also enriched for other mitotic mRNAs, raising the possibility that YTHDC2 may be required to repress the expression of a number of mitotic proteins. While the expression level of *Ccna2* mRNA was slightly higher in *Ythdc2*⁻/⁻ testes compared to wild type, the majority of RNAs bound by YTHDC2 at P12 were expressed at similar levels in wild-type and *Ythdc2* mutant testis, suggesting that YTHDC2 may play a role in translational repression of target RNAs or in regulating the subcellular localization of bound RNAs, rather than affecting RNA stability.

We and others have found that YTHDC2 physically interacts with MEIOC, and the phenotypes of testes and ovaries from *Meioc*⁻/⁻ (*Abby et al., 2016*; *Soh et al., 2017*) and *Ythdc2*⁻/⁻ mice are strikingly similar, suggesting that YTHDC2 and MEIOC work together. As seen in *Meioc* mutants, juvenile testes from *Ythdc2* mutant mice showed markedly decreased expression of many meiotic transcripts. However, our global analysis of YTHDC2 target RNAs by fRIP-Seq at P12 did not show significant enrichment for meiotic prophase transcripts, suggesting that while YTHDC2 may bind a few meiotic transcripts, the role of YTHDC2 in expression or stabilization of many meiotic transcripts may be indirect or result from an inability of *Ythdc2* mutant cells to progress through later spermatocyte stages.

We found that the mouse and human YTHDC2 proteins localize to RNA granules in spermatocytes entering meiotic prophase. Furthermore, mouse YTHDC2 binds the germ granule components MIWI and MSY2, consistent with recent results indicating that mouse YTHDC2 is a component of chromatoid body germ granules (*Meikar et al., 2014*). Germ granules are mainly composed of RNAs and RNA binding proteins, and are key sites for regulating RNA translation and stability (*Voronina et al., 2011*). Along with their role in the piRNA pathway, MIWI and MSY2 have been shown to bind and stabilize and/or store spermiogenic mRNAs in cytoplasmic messenger ribonucleoprotein (mRNP) granules until the mRNAs are required (*Vourekas et al., 2012*; *Yang et al., 2005*; *Yang et al., 2007*). MIWI and MSY2 protein levels increase in late spermatocytes around the pachytene stage, the stage when YTHDC2 protein levels were the highest throughout the cytoplasm. We propose that YTHDC2 may have both an early role in spermatocytes, repressing expression of mitotic targets as the germ cells first enter meiotic prophase, and also a later role in more mature spermatocytes acting with MSY2 and MIWI to stabilize or store transcripts for later use (*Figure 9G*).

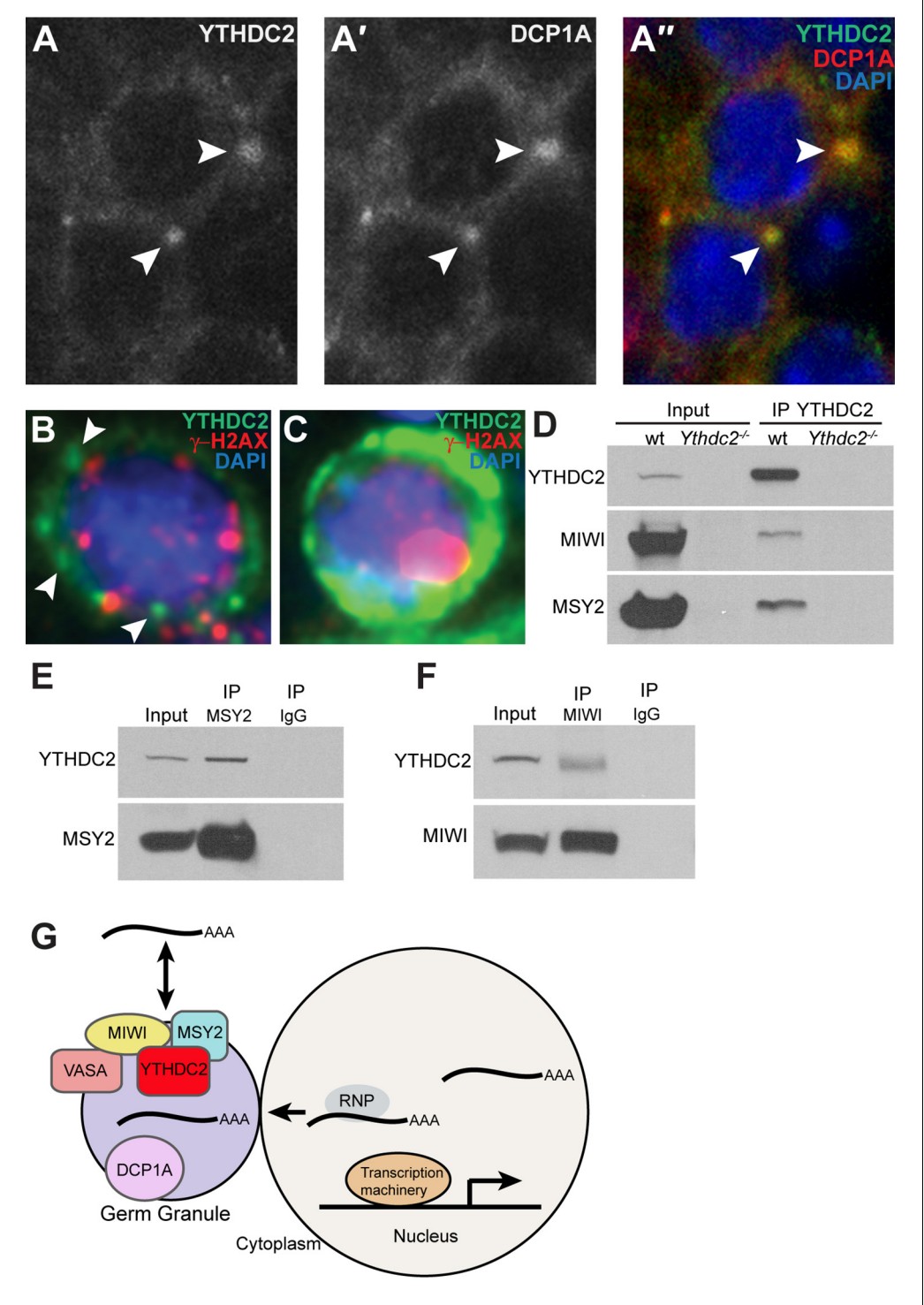

**Figure 9.** YTHDC2 localizes to RNA granules. (**A-A''**) High-magnification immunofluorescence images of spermatocytes from an adult mouse testis cross-section stained for YTHDC2 (green) DCP1A (red) and DAPI (blue). (**B**) High-magnification image of an early spermatocyte from a human testis cross-section stained for YTHDC2 (green), γ-H2AX (red) and DAPI (blue). (**C**) High-magnification image of a late, pachytene-stage spermatocyte from a human testis cross-section stained for YTHDC2 (green), γ-H2AX (red) and DAPI (blue). (**D**) Immunoprecipitation with α-YTHDC2 from adult wild-type (wt) and *Ythdc2⁻/⁻* mouse testis extracts. Western blots probed with α-YTHDC2, α-MIWI, or α-MSY2. (**E**) Immunoprecipitation with α-MSY2 and α-IgG from adult wild-type mouse testis extracts. Western blots probed with α-YTHDC2 and α-MSY2. (**F**) Immunoprecipitation with α-MIWI and α-IgG from

*Figure 9 continued*

adult wild-type mouse testis extracts. Western blots probed with α-YTHDC2 and α-MIWI. (**G**) Model for action of YTHDC2 with germ granule components in spermatocytes. See also ***Figure 9—figure supplement 1***.

DOI: https://doi.org/10.7554/eLife.26116.036

The following figure supplement is available for figure 9:

**Figure supplement 1.** MYBL1 expression in *Ythdc2* mutant testes.

DOI: https://doi.org/10.7554/eLife.26116.037

YTHDC2 also bound several piRNA precursor RNAs including the pachytene piRNA precursor transcript 17-qA3.3–27363.1. Strikingly, the 17-qA3.3–27363.1 piRNA precursor was expressed at a significantly lower level in *Ythdc2* mutant versus wild-type testes. The transcription factor MYBL1 (also known as A-MYB), previously shown to regulate pachytene piRNA precursor transcription (***Li et al., 2013a***), was expressed at similar levels in P12 wild-type and *Ythdc2*[-/-] testes (***Figure 9—figure supplement 1***), suggesting that transcription of piRNA precursor 17-qA3.3–27363.1 may not be affected in *Ythdc2* mutant testes. It is possible that YTHDC2 regulates the stability or processing of the piRNA precursors. Recent studies suggest that pachytene piRNAs may play a role in eliminating a large number of mRNAs, including meiotic transcripts, in late spermatocytes and early spermatids (***Goh et al., 2015***; ***Gou et al., 2014***; ***Watanabe et al., 2015***). Likewise, in *Drosophila* testes, an RNA precursor expressed in primary spermatocytes from repeats on the Y chromosome is processed into piRNAs that down-regulate expression of the X-linked gene *Stellate* (***Bozzetti et al., 1995***; ***Quénerch'du et al., 2016***). One possibility is that YTHDC2 may bind and sequester certain pachytene piRNA precursors in early spermatocytes and block them from being processed into mature piRNAs prematurely.

While loss of function of *Ythdc2* in mice, or *bgcn* in *Drosophila* males prevents germ cells from properly executing the meiotic program, the phenotypes in mice and flies do differ. Mouse male germ cells lacking YTHDC2 function appear to attempt only one additional mitotic division before entering the apoptotic program, while *Drosophila* male germ cells lacking Bgcn function undergo several extra rounds of mitotic division before dying. A previous study suggested that the normal apoptotic pathway that involves a cascade of effector caspases does not appear to be active in *Drosophila* spermatocytes (***Yacobi-Sharon et al., 2013***). In contrast, it is possible that strong apoptotic surveillance pathways in mouse germ cells, especially on the C57BL/6 background, prevent the aberrant male and female germ cells from continuing to proliferate.

YTHDC2 has several RNA-binding domains that are conserved in *Drosophila* Bgcn, including an RNA helicase domain. Mouse and human YTHDC2 also contain a YTH domain not recognizably conserved in *Drosophila* Bgcn. Several recent studies have shown that the YTH domain can bind the dynamic, reversible RNA modification $N^6$-methyadenosine (m$^6$A) (***Dominissini et al., 2012***; ***Patil et al., 2016***; ***Xu et al., 2015***). The m$^6$A modification has been implicated in human disease (***Batista, 2017***), and it is required for driving the transition from pluripotency to differentiation in mammalian embryonic stem cells by marking pluripotency transcripts for rapid turnover (***Batista et al., 2014***; ***Geula et al., 2015***). The m$^6$A modification plays an important role in the mammalian germline, as mice mutant for the RNA demethylase ALKBH5, which reverses the m$^6$A mark, had impaired fertility (***Zheng et al., 2013***). Furthermore, the m$^6$A-modification of RNA occurs only during meiosis in the yeast *Saccharomyces cerevisiae* and is required for efficient meiotic progression and sporulation (***Agarwala et al., 2012***; ***Clancy et al., 2002***; ***Schwartz et al., 2013***), raising the possibility that a deeply conserved mechanism involving m$^6$A-modified RNAs might regulate progression through meiotic prophase from yeast to humans.

The five known YTH-domain m$^6$A reader proteins have different cellular localizations and appear to function through different post-transcriptional control mechanisms to dynamically regulate RNAs and allow fast changes in gene expression. For example, human YTHDF1 has been shown to promote efficient translation of m$^6$A-marked mRNAs in HeLa cells (***Wang et al., 2015***), while binding of YTHDF2 triggers degradation of target m$^6$A-modified mRNAs in P-bodies (***Wang et al., 2014***; ***Zhao et al., 2017***). Thus, a single YTH protein is not expected to bind all m$^6$A-marked mRNAs present in a cell. We found that the RNAs bound by YTHDC2 by fRIP in P12 mouse testes are enriched for the m$^6$A-modification compared to unbound RNAs, suggesting that the m$^6$A modification may

be one mechanism targeting YTHDC2 to RNAs via its YTH domain. The multiple RNA binding domains present in YTHDC2 may allow YTHDC2 to identify and function through different post-transcriptional control mechanisms on various RNA targets as well.

## Materials and methods

### Mice

*Ythdc2* knockout mice were generated according to the scheme depicted in *Figure 2—figure supplement 1A*. Embryonic stem cell lines carrying a targeted gene-trap cassette in the intron between exons 5 and 6 of the *Ythdc2* gene (*Ythdc2$^{tp/tp}$*) were obtained from the European Conditional Mouse Mutagenesis Program (EUCOMM). ES cell lines were mycoplasma negative by PCR screening. Prior to microinjection, ES cell clones were karyotyped and correct targeting was verified by southern blot. Clone HEPD0563_8_G03 (*Ythdc2$^{tp}$*) was injected into albino-C57BL/6 blastocysts. Chimeric males were bred to B6(Cg)-Tyr$^{c-2J}$/J females and the progeny were genotyped by PCR using the following primers: *Ythdc2* 5′arm forward: 5′-CTGAACATGTCTTATCCACAGTGC-3′, *Ythdc2* 3′arm reverse: 5′-CATCATCAAGAAGGTTACAACAGGC-3′, targeting cassette reverse: 5′-CAACGGGTTC TTCTGTTAGTCC-3′. The *Ythdc2* null allele was generated by crossing the *Ythdc2$^{tp/tp}$* mice to B6. FVB-Tg(EIIa-cre)C5379Lmgd/J mice to delete exons 6 and 7 as well as the neo cassette between the LoxP sites. The Cre allele was removed prior to analyzing the mice. Removal of the floxed region was confirmed by PCR using the following primers: *Ythdc2* forward: 5′-CGAGTGCTGCCTTGGATG TGAACC-3′, *Ythdc2* reverse: 5′-GGATTTTGACAGCCTTGAGCCTGGG-3′, targeting cassette reverse: 5′- GAATTATGGCCCACACCAGTGGCG3′. All experiments were approved by the Stanford University Animal Care and Use Committee and performed according to NIH guidelines. All experiments were performed with *Ythdc2* null mutant mice (*Ythdc2$^{-/-}$*) and were maintained on a C57BL/6 background. All experimental wild-type mice used were *Ythdc2$^{+/+}$* unless otherwise indicated.

### Human tissue collection

Human testis sections were obtained from the Fertility Preservation Program of Pittsburgh. The testicular tissue freezing protocol was approved by the Institutional Review Board of the University of Pittsburgh.

### Histology

Mouse testes and ovaries were fixed overnight at room temperature in bouins fixative or 10% formalin and embedded in paraffin. Paraffin-embedded samples were cut to 5–6 μm sections and stained with either hematoxylin and eosin (H and E) or periodic acid-Schiff (PAS).

### Immunofluorescence

For immunofluorescence staining, tissues were fixed with 4% paraformaldehyde overnight at 4°C, embedded in paraffin and cut to 5–6 μm thickness. After rehydrating in an ethanol gradient, heat-mediated antigen retrieval was performed on the sections using sodium citrate buffer (10 mM sodium citrate, 0.05% Tween-20, pH 6.0). Sections were then permeabilized with 0.1% TritonX-100/ PBS (PBST) for 45 min at room temperature, followed by 1 hr incubation with blocking buffer (10% BSA/PBST) at room temperature. Sections were incubated at 4°C with primary antibody overnight. Slides were then incubated with Alexa Fluor-conjugated donkey secondary antibodies (1:500, Molecular Probes, Eugene, OR) at room temperature for 2 hr and then mounted in VECTASHIELD medium with 4′−6-Diamidino-2-phenylindole (DAPI) (Vector Lab Inc., Burlingame, CA). Immunofluorescence controls: For mouse YTHDC2 immunofluorescence experiments (*Figure 1D–E*), YTHDC2 antibody staining in *Ythdc2$^{-/-}$* testes and secondary antibody alone staining were performed as controls alongside wild-type testes. Secondary antibody alone controls were also performed for human YTHDC2 expression experiments (*Figure 1G–I*).

### Antibodies

Rabbit anti-YTHDC2 (1:500, A303-026A, (RRID:AB_10754785), Bethyl laboratories, Montgomery, TX), rabbit anti-STRA8 (1:1,000, Michael Griswold lab), mouse anti-SYCP3 (1:200, clone [Cor 10G11/ 7], (RRID:AB_10678841), Abcam), rabbit anti-REC8 (1:1,000, Richard Schultz lab), rabbit anti-DMC1

(1:250, clone [H-100], (RRID:AB_2277191), Santa Cruz Biotechnology, Santa Cruz, CA), goat anti-DDX4 (1:250, AF2030, (RRID:AB_2277369), R and D Systems, Minneapolis, MN), rabbit anti-phospho-Histone H3 (Thr3) (1:200 clone [JY325], (RRID:AB_310604), Millipore, Burlington, MA), mouse anti-phospho-Histone H2A.X (Ser139) (1:500 clone [JBW301], (RRID:AB_309864), Millipore), rabbit anti-gamma-H2AX (1:250, IHC-00059, (RRID:AB_533402), Bethyl laboratories), rabbit anti-Cyclin D1/Bcl-1 (1:500 clone [SP4], (RRID:AB_720758), Thermo Scientific, Waltham, MA), rabbit anti-Cyclin A2 (1:250 clone [EPR17351], ab181591, Abcam, Cambridge, MA), rabbit anti-MEIOC (1:200 IF, 1:2000 immunoblot, David Page Lab), mouse anti-DCP1A (1:250 clone [3G4], (RRID:AB_1843673) Sigma, St. Louis, MO). Terminal deoxynucleotidyl transferase dUTP nick end labeling (TUNEL) staining was done following manufacturer's instruction (in situ cell death kit TMR, Roche, Basel, Switzerland).

The following antibodies were used for immunoblot: rabbit anti-YTHDC2 (1:1,000, A303-026A, (RRID:AB_10754785), Bethyl laboratories), mouse anti-Actin (1:100,000, clone [C4], (RRID:AB_2223041), Millipore), rabbit anti-MYBL1 (1:1,000, ARP40093_P050, (RRID:AB_2046866), Aviva Systems Biology, San Diego, CA), goat anti-PIWIL1 (1:500, clone [N-17], (RRID:AB_2165430), Santa Cruz Biotechnology), rabbit anti-MIWI (G82) (1:1000, (RRID:AB_2165432), Cell Signaling, Danvers, MA) and mouse anti-MSY2 (1:1,000, clone [A-12], sc-393840, Santa Cruz Biotechnology).

## EdU labeling

Mice were injected intraperitoneally (IP) with 25 µg/g EdU (5-ethynyl-2'-deoxyuridine). Two hours post EdU injection, testes were collected and fixed in 4% paraformaldehyde overnight at 4°C, washed and embedded in paraffin. Sections were subjected to deparaffinization and antigen retrieval as described above. Sections were incubated with primary antibody overnight at 4°C and then Alexa Fluor-488 conjugated secondary antibody for 2 hr at RT. Sections were then incubated with Click-IT EdU Alexa-Fluor-555 reagents according to manufacturer's instructions (Molecular Probes).

## Germ-cell spreads

Chromosome spreads from testes were performed based on a previously published protocol (Peters et al., 1997). Briefly, testes were incubated in a hypotonic extraction buffer pH 8.2 (30 mM Tris-HCL, 50 mM sucrose, 17 mM trisodium citrate dehydrate, 5 mM EDTA, 0.5 mM DTT, 0.5 mM PMSF) for 60 min. Testis tubules were then broken apart in 100 mM sucrose and then spread on a slide dipped in fixation buffer (1% PFA and 0.15% Triton X-100). Slides were incubated overnight in a humidified chamber at room temperature and air-dried. Slides were washed twice in 0.4% Photoflo (Kodak, Rochester, NY) and air-dried.

## Co-immunoprecipitation from testes

Testes were dissected from wild-type and *Ythdc2* mutant testes. Testis extracts were prepared by mechanically disrupting testes in lysis buffer (20 mM Tris-HCL, 135 mM NaCl, 10% glycerol, 10% Nonidet P-40, 5 mM EDTA, 1 mM PMSF and 1x Complete protease inhibitor). Extracts were incubated in lysis buffer for 30 min at 4°C, spun for 10 min, and then precleared with uncoupled Protein A Dynabeads (Invitrogen, Waltham, MA) for 45 min at 4°C. Testis extracts were then added to Protein A Dynabeads chemically crosslinked to rabbit anti-YTHDC2 (A303-025A, (RRID:AB_10752592), Bethyl laboratories), mouse anti-MSY2 (clone [A-12], sc-393840, Santa Cruz Biotechnology), rabbit anti-PIWIL1 (ab12337, (RRID:AB_470241), Abcam), normal mouse IgG (RRID:AB_145840, Millipore) or normal rabbit IgG (RRID:AB_145841, Millipore) with dimethylprimelimidate (DMP, Sigma) (5.4 mg DMP per mL 0.2M triethanolamine) per manufacture's instructions, and incubated for 4 hr at 4°C. Beads were washed twice in lysis buffer and incubated at 70°C for 30 min in elution buffer (1% SDS, 10 mM EDTA, 50 mM Tris pH 8.0, 1 mM PMSF, 1x Complete protease inhibitor) with frequent mixing. Samples were resolved on a 10% SDS-PAGE gel (Bio-Rad, Hercules, CA) and transferred onto polyvinylidene fluoride (PVDF) membranes. Blots were then incubated in primary antibodies overnight at 4°C and then IgG HRP secondary antibodies (1:10,000) for 2 hr at RT and then developed using western lightning ECL detection reagent.

## RNA isolation and qRT-PCR

Total RNA was isolated using TriZol (Life Technologies) and first-strand cDNA was generated with Ready-To-Go You-Prime First-Strand Beads (GE Healthcare, Marlborough, MA) using either oligo-(dT) or random primers. Real-time quantitative PCR (qPCR) reactions were performed on a 7500 Real Time PCR System using the following gene-specific primer sets: *Spo11* 5′-CGTGGCCTCTAG TTCTGAGGT-3′ and 5′-GCTCGATCTGTTGTCTATTGTGA-3′; *Meioc* 5′-AATCTTGGTGCCTAAGTC TATG-3′ and 5′-AGGCTTTATATCCAGCAACTC-3′; *17-qA3.3–27363.1* 5′-GGAAAAGCCTTTCAG-CAGCC-3′ and 5′-ATGGCCCAGACCAGCTACTA-3′; *Ccna2* 5′-CTCAACCCACCAGAGACACT-3′ and 5′- CTGTACAGCATGGACTCCGA −3′; *Rpl7-* 5′- CCCTGAAGACACTTCGAAAGG −3′ and 5′-GCTTTCCTTGCCATCCTAGC-3′; *Hprt-* 5′-TCAGTCAACGGGGGACATAAA 3′ and 5′ GGGGCTG TACTGCTTAACCAG 3′. Relative expression levels were calculated using comparative Ct values after normalizing to control.

## RNA extraction, selection and fragmentation for m⁶A-IP

Total RNA was isolated using TRIzol (Life Technologies), according to manufacturer's instructions. RNA was resuspended in ultrapure water and treated with DNAse I (Ambion) for 30 min at 37°C and subjected to RNA clean up with RNeasy Midi Kit (Qiagen), according to manufacture's protocol. RNA was eluted in ultrapure water. 800 µg of total RNA were subjected to two rounds of poly(A) RNA selection, using the MicroPoly(A) Purist kit, (Life Technologies) according to the manufacturer's protocol. For the second round of poly(A) RNA selection, the eluate of the first round of poly(A) RNA selection was used as starting material. 2.5 µg of poly(A)-selected RNA was incubated for 45 s with pre-warmed Zinc chloride buffer (10 mM $ZnCl_2$, 10 mM Tris-HCl, ph 7.0), to fragment the RNA to ~100 nucleotide long fragments. The reaction was stopped with 0.2M EDTA and immediately placed on ice. The fragmented RNA was recovered using the RNEasy Mini Kit (Qiagen).

## m⁶A immunoprecipitation

25 µl of Dynabeads Protein G (Life Technologies) and 25 µl of Dynabeads Protein A (Life Technologies) were incubated overnight with 2 µg of anti-m⁶A polyclonal antibody (Synaptic Systems) in 1x IPP buffer (150 mM NaCl, 10 mM TRIS-HCL, and 0.1% NP-40). 2.5 µg of fragmented RNA was denatured and incubated with 25 µl of Protein G beads conjugated to anti-m⁶A antibody in 1x IPP buffer with 80 Units of RiboLock RNase Inhibitor (Life Technologies) rotating at 4°C for 3 hr. The beads were washed 2 times with 1x IPP buffer, 2 times with low-salt buffer (50 mM NaCl, 10 mM TRIS-HCL, and 0.1% NP-40), 2 times with high salt buffer (500 mM NaCl, 10 mM TRIS-HCL, and 0.1% NP-40), and 1 time with 1x IPP buffer. RNA was eluted using the RNEasy Mini Kit (Qiagen) and eluted twice in 50 µl of ultrapure water. The RNA was then incubated with 10 µl of Protein A beads conjugated to anti-m⁶A antibody in 1x IPP buffer rotating at 4°C for 3 hr. The beads were washed and RNA eluted as described above.

## Preparation of m⁶A-seq library

Library preparation for high-throughput sequencing was performed following the protocol described in (Flynn et al., 2015). The RNA was treated with 10 Units of T4 PNK (NEB, Ipswich, MA), 1 Unit of FastAP (Thermo Scientific) in 1x PNK buffer with 40 Units of Ribolock RNase Inhibitor for 30 min at 37°C, to repair the 3′ of the RNA. For the ligation of the 3′ end adaptor, 0.8 µM of 3′-end biotin blocked preadenylated adaptor (/5rApp/AGATCGGAAGAGCGGTTCAG/3Biotin/; 5rApp = 5′ pre-adenylation), 6.6 Units of T4 RNA ligase 1, high concentration (NEB), in 1x RNA ligase buffer with 5 mM DTT and 50% PEG8000, were added, and the reaction was incubated at 16°C overnight. To remove excess linker, 30 Units of RecJf (NEB) and 20 Units of 5′ Deadenylase (NEB) in 1x RecJ buffer with Ribolock RNase Inhibitor (Life Technologies) were added to the reaction, and incubated for 1 hr at 37°C. The reaction was cleaned with RNA clean up Zymo Columns. The RNA was lyophilized and resuspended in 14 µL of water with 6 nM of RT primer (/5phos/DDD NNA ACC NNN NAG ATC GGA AGA GCG TCG TGA T/iSp18/GG A TCC/iSp18/TAC TGA ACC GC;/5phos / = 5′ phosphorylation,/iSp18/ =Carbon-18 spacer). The reaction was incubated at 70°C for 5 min and cooled down to 4°C for at least 1 min. 100 Units of TIGR III enzyme and TIGR reaction buffer were added (final concentration of 1x) to the reaction and incubated at 25°C for 30 min. After addition of dNTP (final concentration 1.25 mM), the reaction was incubated at 60°C for 1 hr. 5 Units of RNaseH (Enzymatics)

were added, and the reaction incubated at 37°C for 15 min. 5 µL MyOne C1 (ThermoFisher) were washed in Biotin-Ip-buffer (100 mM Tris-HCl pH 7.0, 1M NaCl, 0.1% Tween and 1 mM EDTA) and added to each sample. Reactions were rotated for 30 min at room temp. Beads were washed 2x in NT2 buffer (50 mM Tris-HCl pH 7.0, 150 mM NaCl, 1 mM $MgCl_2$ and 0.05% NP-40) and resuspended in 10 µL of cDNA elution mix (0.1 uM short primer (5'-CTGAACCGCTCTTCCGATCT-3'), 3.75 mM $Mm^{2+}$) and incubated 95°C for 10 min and slowly cooled down to 60°C. CircLigase II (Epicenter) was added with CircLigase II buffer (final concentration 1x) and the reaction incubated at 60°C for 90 min. 30 µL of SPRI beads (Beckman Coulter, Brea, CA) were added to the reaction and the cDNA precipitated with 1.6 volumes of Isopropanol. After 10 min incubation (with shaking), the beads were washed with 85% Ethanol, and the cDNA eluted in 10 µl of ultrapure water. The cDNA was amplified in a PCR reaction with 1x Phusion HF Master Mix (NEB) including the primers: (P3 Solexa: 5'-CAA GCA GAA GAC GGC ATA CGA GAT CGG TCT CGG CAT TCC TGC TGA ACC GCT CTT CCG ATC T-3'; P5 Solexa: 5'-AAT GAT ACG GCG ACC ACC GAG ATC TAC ACT CTT TCC CTA CAC GAC GCT CTT CCG ATC T-3'), and individual reactions stopped at the end of the extension step once reaching a fluorescence value of 8000 units on a Mx3000 qPCR System (Agilent, Santa Clara, CA). DNA was recovered with SPRI beads (4 uL of isopropanol and 36 µL of beads) and eluted in 10 µL of water. The DNA was amplified for three cycles and clean with a Zymo DNA clean-up reaction. The PCR products were resolved on a PAGE gel and the products in the 150 to 250 nucleotide range excised. DNA was eluted from gel overnight in Elution buffer (10 mM Tris pH 7.5, 500 mM NaCl, 1 mM EDTA and 0.1% SDS). One microliter of each sample was used for quantification with Kapa Library quantification (Kapa Biosystems) and then sent for deep sequencing on the Illumina NextSeq for $1 \times 75$ bp cycle run.

## $m^6A$-IP seq Analysis

Libraries were separated by barcode, and perfectly matching reads collapsed. Reads were first mapped to mouse ribosomal RNA. Reads aligning to the rRNA were discarded, and remaining reads mapped to the mouse genome (mm9 assembly) using STAR (*Dobin et al., 2013*). A nonredundant mm9 transcriptome was assembled from UCSC RefSeq genes, UCSC genes, and predictions from (*Ulitsky et al., 2011*) and (*Guttman et al., 2011*). We performed the search for enriched peaks by scanning each gene using 100-nucleotide sliding windows and calculating an enrichment score for each sliding window (*Dominissini et al., 2012*). HOMER software package (*Heinz et al., 2010*) was used for de novo discovery of the methylation motif. Metagene was generated using the MetaPlotR package (*Olarerin-George and Jaffrey, 2017*).

## RNA expression analysis

Libraries were separated by barcode, matching reads were collapsed and barcodes removed. For all libraries, single-end RNA-Seq reads were mapped to the mouse (mm9 assembly) genome using STAR (*Dobin et al., 2013*). A non-redundant mm9 transcriptome was assembled from UCSC RefSeq genes, UCSC genes, and predictions from (*Guttman et al., 2011*; *Ulitsky et al., 2011*) and piRNA precursor annotation from (*Li et al., 2013a*). Reads for each transcript were extracted using HTSeq (RRID:SCR_005514) (*Anders et al., 2015*). Differential gene expression was calculated with DESeq2 (*Love et al., 2014*). Datasets from the NCBI GEO database (*Edgar et al., 2002*): GEO accession GSE69946 (*Sin et al., 2015*), GEO accession GSE43717 (*Soumillon et al., 2013*) and GEO accession GSE44346 (*Margolin et al., 2014*) were processed as described above. To make UCSC read coverage tracks, the read coverage at each single nucleotide was normalized to library size.

## fRIP

Formaldehyde RNA immunoprecipitation (fRIP) was performed as described in (*G Hendrickson et al., 2016*) with a few modifications. P12 testes were cross-linked in 0.1% formaldehyde at room temperature for 10 min. Crosslinking reaction was quenched by adding glycine to a final concentration of 125 mM and incubating for 5 min at room temperature. Cross-linked testes were washed 2x in 4°C PBS, flash frozen and stored at −80°C. Frozen testes were resuspended in RIPA lysis buffer (50 mM Tris-HCl pH8.0, 150 mM KCl, 0.1% SDS, 1% Triton-X-100, 5 mM EDTA, 0.5% sodium deoxycholate, 0.5 mM DTT, 1x Complete protease inhibitor, and 100 U/mL RNasOUT). Testes were mechanically disrupted in lysis buffer and incubated for 15 min at 4°C then sonicated using a

Bioruptor (1 cycle, 30 s on/off, medium setting at 4°C). After lysis, lysates were centrifuged for 10 min at max speed at 4°C. The supernatant was collected and diluted with an equal volume of binding/wash buffer (25 mM Tris-HCl pH7.5, 150 mM KCl, 5 mM EDTA, 0.5% NP-40, 0.5 mM DTT, 1x Complete protease inhibitor, and 100 U/mL RNasOUT). Lysates were precleared by incubation with uncoupled Protein A Dynabeads (Invitrogen) for 1 hr at 4°C. Following preclearing, 10% of the lysate was saved for input and stored at 4°C, and the remaining lysate was added to Protein A Dynabeads chemically crosslinked to rabbit anti-YTHDC2 (A303-025A, (RRID:AB_10752592) Bethyl laboratories) or normal rabbit IgG (RRID:AB_145841, Millipore) with dimethylprimelimidate (DMP, Sigma) (5.4 mg DMP per mL 0.2M triethanolamine) per manufacture's instructions, and incubated for 4 hr at 4°C. After incubation, samples were washed 4x with binding/wash buffer for 10 min at 4°C.

## fRIP RNA purification and library construction

Beads and Input samples for two biological replicates of both wild type and $Ythdc2^{-/-}$ were resuspended in PK buffer (10 mM Tris pH 7.0, 100 mM NaCl, 1 mM EDTA, 0.5% SDS), 5 μL Proteinase K and 1 μL RNasOUT (100 μL total volume). Samples were incubated for 1 hr at 42°C and then 1 hr at 55°C. RNA was isolated using TRIzol (Life Technologies), according to manufacturer's instructions. 1 μL GlycoBlue and 750 μL isopropanol were added to the aqueous phase and incubated at −20°C. Samples were centrifuged for 40 min at max speed at 4°C and washed 2x in 75% cold ethanol. The RNA was resuspended in 10 μL ultrapure water. Ribosomal RNA was removed from the Input and fRIP samples using the RiboMinus Eukaryote System v2 Kit (Ambion) following manufacture's instructions. Library preparation for high-throughput sequencing was performed using a TruSeq RNA Sample preparation Kit (Illumina, San Diego, CA) per manufacture's instruction, except RNA was not fragmented with FPF solution. One microliter of each sample was used for quantification with Kapa Library quantification (Kapa Biosystems) and then sent for deep sequencing on the Illumina NextSeq for 2 × 75 bp cycle run.

## fRIP-Seq analysis

For all libraries, pair-end RNA-Seq reads were first mapped to mouse ribosomal RNA. Reads aligning to the rRNA were discarded, and remaining reads mapped to the mouse genome (mm9 assembly) using STAR (*Dobin et al., 2013*). Reads for each transcript were extracted using HTSeq (RRID: SCR_005514) (*Anders et al., 2015*). Enrichment was calculated with multifactor analysis using DESeq2 (*Love et al., 2014*).

## Statistics

A Chi-squared with Yates correction (two-tailed) test was used to compare the overlap between the fRIP-Seq enriched transcripts and $m^6A$-modified transcripts in P12 testes as well as the overlap between the under-expressed transcripts in P12 and P14 *Ythdc2* mutants and transcripts up-regulated in wild-type spermatocytes. Biological replicates were performed for RNA-Seq, $m^6A$-IP and fRIP-Seq experiments, samples for each biological replicate were collected from different animals. Samples Sizes: For phenotypic analysis, a sample size of 3 or greater was used, which is similar to what is generally utilized in the field. After data on three replicates was collected, effect size calculations were performed to determine if more replicates would be necessary. Each animal was considered a biological replicate. Sample sizes (n) for each experiment are given in the respective figure legends. Representative images are shown for all histology and immunofluorescence experiments.

## Data availability

All of the data sets have been deposited in the Gene Expression Omnibus (GEO) under accession number GSE93567. Link for access to GSE93567: https://www.ncbi.nlm.nih.gov/geo/query/acc.cgi?token=ufuryeoobvedrwx&acc=GSE93567.

## Acknowledgements

We thank Jeanine Van Nostrand and Manfred Baetscher for valuable experimental help and advice; members of the Fuller Lab for helpful discussions; Pauline Chu from the Stanford Comparative Medicine Animal Histology Services for help preparing histological sections; and the Stanford Functional

Genomics Facility for high-throughput sequencing. We thank Michael Griswold, Richard Schultz and David Page for providing antibodies, and Kyle Orwig for providing human testis samples.

## Additional information

### Funding

| Funder | Grant reference number | Author |
| --- | --- | --- |
| National Institutes of Health | NIH T32 HD007249 | Alexis S Bailey |
| The Lalor Foundation | Postdoctoral Fellowship | Alexis S Bailey |
| National Cancer Institute | Intramural Research Program of the NIH | Pedro J Batista |
| National Institutes of Health | NIH R01 HG004361 | Howard Y Chang Pedro J Batista |
| Reed-Hodgson Professorship | | Margaret T Fuller |
| National Institutes of Health | NIH P50 HD068158 | Margaret T Fuller |

The funders had no role in study design, data collection and interpretation, or the decision to submit the work for publication.

### Author contributions

Alexis S Bailey, Conceptualization, Formal analysis, Validation, Investigation, Visualization, Methodology, Writing—original draft, Project administration; Pedro J Batista, Software, Formal analysis, Validation, Investigation, Visualization, Methodology, Writing—review and editing; Rebecca S Gold, Validation, Investigation, Visualization; Y Grace Chen, Validation, Investigation, Methodology; Dirk G de Rooij, Formal analysis, Supervision, Writing—review and editing; Howard Y Chang, Resources, Supervision, Funding acquisition, Writing—review and editing; Margaret T Fuller, Conceptualization, Resources, Supervision, Funding acquisition, Methodology, Writing—review and editing

### Author ORCIDs

Alexis S Bailey http://orcid.org/0000-0001-8838-8625
Pedro J Batista http://orcid.org/0000-0003-1055-2354
Rebecca S Gold http://orcid.org/0000-0002-2408-7716
Y Grace Chen http://orcid.org/0000-0003-3574-5734
Dirk G de Rooij http://orcid.org/0000-0003-3932-4419
Howard Y Chang http://orcid.org/0000-0002-9459-4393
Margaret T Fuller http://orcid.org/0000-0002-3804-4987

### Ethics

Human subjects: Human testis sections were obtained from the Fertility Preservation Program of Pittsburgh. Informed consent was obtained and the testicular tissue freezing protocol was approved by the Institutional Review Board of the University of Pittsburgh.

Animal experimentation: This study was performed in strict accordance with the recommendations in the Guide for the Care and Use of Laboratory Animals of the National Institutes of Health (NIH). All of the experiments were approved by the Stanford University Animal Care and Use Committee (IACUC), protocol (#21656).

### Decision letter and Author response

Decision letter https://doi.org/10.7554/eLife.26116.049
Author response https://doi.org/10.7554/eLife.26116.050

## Additional files

### Supplementary files

• Supplementary file 1. Breeding data for *Ythdc2* heterozygous (*Ythdc2*$^{+/-}$) mice showing that *Ythdc2* null mutant mice are born at the expected Mendelian ratio.
DOI: https://doi.org/10.7554/eLife.26116.038

• Supplementary file 2. Breeding data for *Ythdc2* homozygous mutant (*Ythdc2*$^{-/-}$) mice showing that *Ythdc2*$^{-/-}$ mice are infertile.
DOI: https://doi.org/10.7554/eLife.26116.039

• Transparent reporting form
DOI: https://doi.org/10.7554/eLife.26116.040

### Major datasets

The following dataset was generated:

| Author(s) | Year | Dataset title | Dataset URL | Database, license, and accessibility information |
|---|---|---|---|---|
| Bailey AS, Batista PJ, Gold RS, Chen YG, de Rooij DG, Chang HY, Fuller MT | 2017 | Regulation of the transition from mitosis to meiosis by the conserved RNA-helicase YTHDC2/BGCN | https://www.ncbi.nlm.nih.gov/geo/query/acc.cgi?&acc=GSE93567 | NCBI Gene Expression Omnibus (accession no: GSE93567) |

The following previously published datasets were used:

| Author(s) | Year | Dataset title | Dataset URL | Database, license, and accessibility information |
|---|---|---|---|---|
| Hosu S, Kartashov AV, Hasegawa K, Barski A, Namekawa SH | 2015 | Poised chromatin and bivalent domains facilitate the mitosis-to-meiosis transition in the male germline | https://www.ncbi.nlm.nih.gov/geo/query/acc.cgi?&acc=GSE69946 | NCBI Gene Expression Omnibus (accession no: GSE69946) |
| Soumillon M, Necsulea A, Weier M, Brawand D, Zhang X, Gu H, Barthès P, Kokkinaki M, Nef S, Gnirke A, Dym M, de Massy B, Mikkelsen TS, Kaessmann H | 2013 | Cellular source and mechanisms of high transcriptome complexity in the mammalian testis (RNA-Seq cells) | https://www.ncbi.nlm.nih.gov/geo/query/acc.cgi?&acc=GSE43717 | NCBI Gene Expression Omnibus (accession no: GSE43717) |
| Margolin G, Khil PP, Kim J, Bellani MA, Camerini-Otero RD | 2014 | RNA-Seq and RNA Polymerase II ChIP-Seq of mouse spermatogenesis | https://www.ncbi.nlm.nih.gov/geo/query/acc.cgi?&acc=GSE44346 | NCBI Gene Expression Omnibus (accession no: GSE44346) |

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
