## [Decision Letter]

Thank you for submitting your article "The conserved RNA helicase YTHDC2 regulates the transition from proliferation to differentiation in the germline" for consideration by *eLife*. Your article has been reviewed by three peer reviewers, and the evaluation has been overseen by a Reviewing Editor and Didier Stainier as the Senior Editor. The reviewers have opted to remain anonymous.

The reviewers have discussed the reviews with one another and the Reviewing Editor has drafted this decision to help you prepare a revised submission.

This paper identifies YTHDC2 as a mammalian ortholog of Bgcn, a translational repressor involved in the differentiation and meiotic onset of germ line stem cells in *Drosophila*. This paper primarily describes the knockout phenotype of YTHDC2 in the development of mouse germ cells. The authors found that mutants of Ythdc2 in both sexes exhibit aberrant germ cell development and infertility, with the abnormalities most visible at a critical stage of mitosis-meiosis transition. Based on knockout phenotypes and analyses of YTHDC2-bound transcripts, the authors concluded that YTHDC2 facilitates meiotic onset by directly clearing mitotic transcripts.

The reviewers agreed that this is an interesting manuscript that is informative about an important but poorly understood developmental process: the transition in the germline from mitosis to meiosis. It is, however, not currently appropriate for publication in *eLife* and further work will be needed, primarily to clarify and better describe the phenotype, but also to reconsider some of the interpretation.

The phenotypic description needs to be improved and more quantified. Assuming that the observations that are described have already been done in more than a single sample, this should not represent a major hurdle. This was agreed by all reviewers to be an important issue for much of the data presented.

Part of this more quantitative approach needs to include rigorous developmental staging of samples, to ensure that comparisons between wild-type and mutant samples are appropriate. For example, the authors need to ensure that the Ccna2 expression that they see in the Ythdc2 mutants represents abnormal expression of Ccna2 in a stage of spermatogenesis that does not normally express Ccna2 (leptotene) rather than wild-type spermatocytes progressing further in spermatogenesis than the Ythdc2 mutants resulting in different stages of spermatogenesis being compared (leptotene wild-type vs pre-leptotene mutant in the current images).

The authors concluded that the YTHDC2 represses mitotic program and thus facilitate the transition to meiotic program but the reviewers were not convinced that this is the only interpretation of the data. The transcriptome data from mutant testes do not show strong evidence of mitotic transcript clearance (i.e., only 27 transcripts are upregulated, and only Ccna2 appears to be related to cell cycle). Second, most transcripts bound by YTHDC2 (722 out of 758) do not show any differences in the transcript levels between wild-type and mutant testes. It could mean that YTHDC2-containing RNA granules simply act as reservoirs, and the fate of those transcripts is determined by the actions of other regulatory factors. It seems premature to conclude that the main function of YTHDC2 is through clearance of target transcripts. Translational control may also be a relevant mechanism.

More specific points

1) The authors describe Cyclin A2 as a mitotic cyclin, and use cyclin A2 to demonstrate failure to transition from mitosis to meiosis in Ythcd2 mutants. However, cyclin A2 is reported to be expressed at the RNA and protein level in preleptotene spermatocytes undergoing meiotic S phase (Ravnik and Wolgemuth, 1996). Could the authors please define which cell types the Abcam antibody to cyclin A2 is staining in their immunofluorescence assay. If cyclin A2 is expressed in the preleptotene stage it should not be described as mitotic/mitotic-specific, and the Ythcd2 phenotype may be more about regulating progression through meiosis (meiotic S phase to meiotic prophase transition) than meiosis vs mitosis.

2) It will be key for the authors to accurately determine whether the cells in Figure 6 that co-express SYCP3 and cyclin A2 represent a normal stage in spermatogenesis or an aberrant combination of meiotic and mitotic markers as they suggest. The SYCP3 patterns in the Ythdc2 mutant cells in Figure 6 look like preleptotene – a stage that is reported to normally express cyclin A2 (Ravnik and Wolgemuth, 1996). The SYCP3 pattern in the wild-type control cells shown in Figure 6 suggests that these cells are probably in zygotene, a stage that does not normally express cyclin A2 (Ravnik and Wolgemuth, 1996). Please can the authors show developmentally matched wild-type and mutant spermatocytes in the preleptotene stage of meiosis for this figure, and quantify the number of preleptotene cells that express/do not express cyclin A2 for each genotype.

3) The comparison between wild-type and Ythdc2 mutant spermatocytes in Figure 4 could be better. The two overlapping nuclei in the wild-type image are in late zygotene/early pachytene, the mutant spermatocyte is in leptotene. Please show a wild-type spermatocyte at a similar stage of leptotene (similar extent of axial element formation) as the mutant spermatocyte.

4) The fRIP bioinformatic analysis would benefit from having candidate targets experimentally validated by qRT-PCR on fRIP'd material.

5) The co-IP between Ythdc2, Miwi and Msy2 would benefit from validation by co-IP'ing Ythdc2 with Miwi and Msy2 antibodies as the use of Ythdc2 mutant input lysate does not control for cross-reaction by the Ythdc2 antibodies.

6) The m6A analysis is preliminary and its conclusions are not fully supported by the data. The data shows that mRNAs that are modified by m6A are also enriched in the RNAs bound by YTHDC2. But the authors have not shown that the mRNA molecules bound by YTHDC2 contain m6A in support of their suggestion that YTHDC2 selects target RNAs through m6A (Discussion paragraph four). The authors need to show that m6A-containing RNAs are present in the YTHDC2 fRIP for this logic to hold. The enrichment itself is fairly modest, and given that only 603 m6A-modified mRNAs are bound by YTHDC2 but 7,105 m6A-modified mRNAs are not, the suggestion that YTHDC2 selects target RNAs through m6A (Results final sentence) is not compelling.

[Editors' note: further revisions were requested prior to acceptance, as described below.]

Thank you for resubmitting your work entitled "The conserved RNA helicase YTHDC2 regulates the transition from proliferation to differentiation in the germline" for further consideration at *eLife*. Your revised article has been favorably evaluated by Didier Stainier (Senior editor), a Reviewing editor, and one reviewer.

The manuscript has clearly been improved. This is a nice paper that reports important evidence that YTHDC2 plays an important role in regulating progression through meiosis in mouse germ cells. The images and data in the revision are good quality and clear, but there are some remaining issues that need to be addressed before acceptance, as outlined below. The key remaining issue is one of quantification of the data, largely in line with the transparent reporting guidelines. I assume that you will have most of what's needed, so fixing this should not be a huge job.

*Reviewer #3:*

This is an interesting and comprehensive study that reports a role for Ythdc2, the mouse ortholog of *Drosophila* Bgcn, in promoting meiotic progression. The authors use mouse genetics, histological and molecular phenotyping, transcriptomics and RNA-IP to show that Ythdc2 binds to mitotic RNAs and regulates their expression to ensure that they do not perturb meiosis. The authors show that Ythdc2 mutant germ cells ectopically express cyclin A2, prematurely exit meiotic prophase, and undergo cell death. These findings represents a significant advance in our understanding of how meiotic progression is regulated in mammals.

The authors have largely revised the text to take into account most of the reviewers comments. The revised data showing cyclin A2 mis-expression in Ythdc2 mutant spermatocytes are convincing, and the fRIP-PCR confirmations reinforce the findings from the fRIP-seq.

---

## [Author Response]

The reviewers agreed that this is an interesting manuscript that is informative about an important but poorly understood developmental process: the transition in the germline from mitosis to meiosis. It is, however, not currently appropriate for publication in eLife and further work will be needed, primarily to clarify and better describe the phenotype, but also to reconsider some of the interpretation.The phenotypic description needs to be improved and more quantified. Assuming that the observations that are described have already been done in more than a single sample, this should not represent a major hurdle. This was agreed by all reviewers to be an important issue for much of the data presented.Part of this more quantitative approach needs to include rigorous developmental staging of samples, to ensure that comparisons between wild type and mutant samples are appropriate. For example, the authors need to ensure that the Ccna2 expression that they see in the Ythdc2 mutants represents abnormal expression of Ccna2 in a stage of spermatogenesis that does not normally express Ccna2 (leptotene) rather than wild-type spermatocytes progressing further in spermatogenesis than the Ythdc2 mutants resulting in different stages of spermatogenesis being compared (leptotene wild-type vs pre-leptotene mutant in the current images).

We agree with the reviewers that it is critical to compare similar developmental stages of spermatogenesis in order to make accurate comparisons between *Ythdc2* mutants and wild-type controls. We have carefully compared the levels of Cyclin A2 protein expression in wild-type and *Ythdc2^-/-^* leptotene stage spermatocytes as requested, providing new images as well as quantification in a new Figure 7 as well as in the Results section (see below under point #2 for details).

The authors concluded that the YTHDC2 represses mitotic program and thus facilitate the transition to meiotic program but the reviewers were not convinced that this is the only interpretation of the data. The transcriptome data from mutant testes do not show strong evidence of mitotic transcript clearance (i.e., only 27 transcripts are upregulated, and only Ccna2 appears to be related to cell cycle). Second, most transcripts bound by YTHDC2 (722 out of 758) do not show any differences in the transcript levels between wild type and mutant testes. It could mean that YTHDC2-containing RNA granules simply act as reservoirs, and the fate of those transcripts is determined by the actions of other regulatory factors. It seems premature to conclude that the main function of YTHDC2 is through clearance of target transcripts. Translational control may also be a relevant mechanism.

We agree with the reviewers that YTHDC2 may normally silence target transcripts by translational control, so it is premature to conclude that the main function of YTHDC2 is through clearance of target transcripts. Indeed, as the reviewers point out, for many of the target transcripts detected as bound by YTHDC2 by our fRIP analysis, expression levels did not change between wild-type and *Ythdc2* mutant testes. We represent these data in new volcano plots in Figure 8—figure supplement 1. We have reworded our conclusions to indicate that both RNA degradation and translational control are potential mechanisms.

More specific points1) The authors describe Cyclin A2 as a mitotic cyclin, and use cyclin A2 to demonstrate failure to transition from mitosis to meiosis in Ythcd2 mutants. However, cyclin A2 is reported to be expressed at the RNA and protein level in preleptotene spermatocytes undergoing meiotic S phase (Ravnik and Wolgemuth, 1996). Could the authors please define which cell types the Abcam antibody to cyclin A2 is staining in their immunofluorescence assay. If cyclin A2 is expressed in the preleptotene stage it should not be described as mitotic/mitotic-specific, and the Ythcd2 phenotype may be more about regulating progression through meiosis (meiotic S phase to meiotic prophase transition) than meiosis vs mitosis.

See combined response for specific points 1 and 2 below.

2) It will be key for the authors to accurately determine whether the cells in Figure 6 that co-express SYCP3 and cyclin A2 represent a normal stage in spermatogenesis or an aberrant combination of meiotic and mitotic markers as they suggest. The SYCP3 patterns in the Ythdc2 mutant cells in Figure 6 look like preleptotene – a stage that is reported to normally express cyclin A2 (Ravnik and Wolgemuth, 1996). The SYCP3 pattern in the wild-type control cells shown in Figure 6 suggests that these cells are probably in zygotene, a stage that does not normally express cyclin A2 (Ravnik and Wolgemuth, 1996). Please can the authors show developmentally matched wild-type and mutant spermatocytes in the preleptotene stage of meiosis for this figure, and quantify the number of preleptotene cells that express/do not express cyclin A2 for each genotype.

Although Ravnik and Wolgemuth, 1996, reported that Cyclin A2 is expressed in mitotic spermatogonia and preleptotene spermatocytes, a subsequent paper by Wolgemuth et al. suggested that Cyclin A2 is highly expressed in differentiated A1 to B spermatogonia, but is down-regulated in preleptotene spermatocytes (Int. J. Dev. Biol., 2013). We revisited this analysis with the current antibody reagents and in the C57BL/6 mouse background on which our *Ythdc2* mutants were made.

Based on our Cyclin A2 and SYCP3 immunofluorescence staining, we found that in wild-type testes, Cyclin A2 protein expression was high in mitotic spermatogonia and decreased in preleptotene spermatocytes (new Figure 7—figure supplement 1). *Ythdc2* mutant testis cross-sections stained in parallel showed higher levels of Cyclin A2 expression in preleptotene spermatocytes (Figure 7—figure supplement 1) compared to wild-type. We then extended our analysis to leptotene-stage spermatocytes, which have entered meiotic prophase, taking care to assess developmentally-matched wild-type and *Ythdc2* mutant cells where both appeared to be in leptotene based on the SYCP3 pattern. In wild-type, Cyclin A2 is normally fully down-regulated in leptotene cells (Figure 7), in agreement with Ravnik and Wolgemuth, 1996, and Wolgemuth et al., 2013. However, in *Ythdc2* mutant testes, Cyclin A2 protein remained highly expressed in leptotene spermatocytes (Figure 7). New Figure 7 now provides images of wild-type and *Ythdc2* mutant leptotene spermatocytes at both low and high magnification, as well as quantification of the number of wild-type and *Ythdc2* mutant leptotene cells that express Cyclin A2. All of the preleptotene and leptotene images were taken from the same staining, and the wild-type and *Ythdc2* mutant testis sections were stained in parallel to allow proper comparison of Cyclin A2 expression levels. In addition, we have clarified our wording throughout the text to indicate that *Ythdc2* mutant germ cells do enter meiosis but fail to properly progress through meiotic prophase.

3) The comparison between wild-type and Ythdc2 mutant spermatocytes in Figure 4 could be better. The two overlapping nuclei in the wild-type image are in late zygotene/early pachytene, the mutant spermatocyte is in leptotene. Please show a wild-type spermatocyte at a similar stage of leptotene (similar extent of axial element formation) as the mutant spermatocyte.

We have replaced the wild-type spermatocyte image (now Figure 4) to show a similar stage of leptotene to match the *Ythdc2* mutant image.

4) The fRIP bioinformatic analysis would benefit from having candidate targets experimentally validated by qRT-PCR on fRIP'd material.

We have validated multiple targets by independent fRIP followed by qRT-PCR, including *Ccna2, Meioc* and the piRNA precursor *17qA3.3-27363* (Figure 8). In addition to these positive fRIP targets, we also included several negative controls, which our fRIP-Seq analysis indicated were not bound by YTHDC2. The fRIP-qRT-PCR confirmed our fRIP-Seq results for both the RNAs bound by YTHDC2 and the RNAs not selectively enriched by IP of YTHDC2.

5) The co-IP between Ythdc2, Miwi and Msy2 would benefit from validation by co-IP'ing Ythdc2 with Miwi and Msy2 antibodies as the use of Ythdc2 mutant input lysate does not control for cross-reaction by the Ythdc2 antibodies.

We now show reciprocal co-IPs of YTHDC2 with both anti-MIWI and anti-MSY2. We also provide IgG controls alongside the reciprocal IPs (Figure 9).

6) The m6A analysis is preliminary and its conclusions are not fully supported by the data. The data shows that mRNAs that are modified by m6A are also enriched in the RNAs bound by YTHDC2. But the authors have not shown that the mRNA molecules bound by YTHDC2 contain m6A in support of their suggestion that YTHDC2 selects target RNAs through m6A (Discussion paragraph four). The authors need to show that m6A-containing RNAs are present in the YTHDC2 fRIP for this logic to hold. The enrichment itself is fairly modest, and given that only 603 m6A-modified mRNAs are bound by YTHDC2 but 7,105 m6A-modified mRNAs are not, the suggestion that YTHDC2 selects target RNAs through m6A (Results final sentence) is not compelling.

We agree with the reviewers that the data presented do not prove that YTHDC2 binds m^6^A modified RNAs. However, the experiments requested are not feasible in vivo with current techniques. We have reworded the text in the Introduction, Results and Discussion sections to state that our results agree with the hypothesis that YTHDC2 binds m^6^A modified RNAs, as shown in previous publications, but further work will have to be performed to dissect how much the YTH domain (which binds m^6^A modified RNAs) versus the other predicted RNA binding domains of YTHDC2 contribute to target selection by YTHDC2. We also now mention in the Discussion section that since there are five different YTH domain family members, which have different cellular localization and different effects on RNAs, we do not expect a single YTH protein to bind all m^6^A-marked RNAs in a cell.

[Editors' note: further revisions were requested prior to acceptance, as described below.]

The manuscript has clearly been improved. This is a nice paper that reports important evidence that YTHDC2 plays an important role in regulating progression through meiosis in mouse germ cells. The images and data in the revision are good quality and clear, but there are some remaining issues that need to be addressed before acceptance, as outlined below. The key remaining issue is one of quantification of the data, largely in line with the transparent reporting guidelines. I assume that you will have most of what's needed, so fixing this should not be a huge job.

We have added new phenotypic quantification throughout the manuscript in response to the reviewers’ requests and have addressed all of the reviewers’ specific points listed below. In addition, we have included in the text how sample sizes were determined and provide the sample sizes for each experiment in the figure legends. The additional quantifications support our conclusions and strengthen our manuscript and we thank the reviewers for their suggestions.